# Lovász Principle for Unsupervised Graph Representation Learning

**Ziheng Sun**[1,2]     **Chris Ding**[1]     **Jicong Fan** [1,2*]
[1]School of Data Science, The Chinese University of Hong Kong, Shenzhen, China
[2]Shenzhen Research Institute of Big Data, Shenzhen, China
zihengsun@link.cuhk.edu.cn     {chrisding,fanjicong}@cuhk.edu.cn

## Abstract

This paper focuses on graph-level representation learning that aims to represent graphs as vectors that can be directly utilized in downstream tasks such as graph classification. We propose a novel graph-level representation learning principle called Lovász principle, which is motivated by the Lovász number in graph theory. The Lovász number of a graph is a real number that is an upper bound for graph Shannon capacity and is strongly connected with various global characteristics of the graph. Specifically, we show that the handle vector for computing the Lovász number is potentially a suitable choice for graph representation, as it captures a graph's global properties, though a direct application of the handle vector is difficult and problematic. We propose to use neural networks to address the problems and hence provide the Lovász principle. Moreover, we propose an enhanced Lovász principle that is able to exploit the subgraph Lovász numbers directly and efficiently. The experiments demonstrate that our Lovász principles achieve competitive performance compared to the baselines in unsupervised and semi-supervised graph-level representation learning tasks. The code of our Lovász principles is publicly available on GitHub[†].

## 1   Introduction

Graphs, such as chemical compounds, protein structures, and social networks, are non-Euclidean data that represent the relationships between entities. There have been a large number of previous works studying many aspects of graphs, including mutagenicity prediction of chemical compounds [Debnath *et al.*, 1991; Kriege and Mutzel, 2012], protein structure prediction [Borgwardt *et al.*, 2005], and community analysis of social networks [Yanardag and Vishwanathan, 2015].

Graph-based learning problems can be organized into two categories: node-level learning and graph-level learning. In this paper, we will only focus on graph-level learning. It is known that in graph-level learning, one fundamental task or step is to measure the distance or similarity between graphs. An important class of methods comparing graphs is graph kernel and many graph kernels have been proposed in the past decades [Siglidis *et al.*, 2020]. For instance, random walk kernels [Gärtner *et al.*, 2003] are the most widely-used and well-studied graph kernel family, which measure the graph similarity by counting the common random walks between graphs. The Weisfeiler-Lehman [Weisfeiler and Leman, 1968] family kernels are based on node label reassignment. Most graph kernels extract the similarity information between graphs by sampling sub-structures of graphs such as walks or reassigning the attributes of nodes with their neighborhoods. Note that graph kernels are implicit graph representation methods and hence their flexibilities are not high. In addition, the time and space complexities are quadratic with the number of graphs.

---

[*]Corresponding author
[†]https://github.com/SunZiheng0/Lovasz-Principle

Graph representation learning aims to convert data with graph structure into vector representations that can be applied to various downstream tasks, such as graph clustering and classification. Many studies have been conducted on graph-level representation learning, and some of them use neural message-passing algorithms [Kipf *et al.*, 2018; Xie and Grossman, 2018; Gilmer *et al.*, 2017]. For instance, the InfoGraph proposed by [Sun *et al.*, 2019] achieves graph-level representations by maximizing the mutual information between the graph-level representation and the node-level representations. Graph contrastive learning (GraphCL) [You *et al.*, 2020] and adversarial graph contrastive learning (AD-GCL) [Suresh *et al.*, 2021] obtain graph-level representations by training graph neural networks (GNNs) to maximize the correspondence between the same graph's representations in its various augmented forms. JOint Augmentation Optimization (JOAO) [You *et al.*, 2021] is a framework that automatically and adaptively selects data augmentations for GraphCL on specific graph data, using a unified bi-level min-max optimization approach. Automated Graph Contrastive Learning (AutoGCL) [Yin *et al.*, 2022] uses learnable graph view generators and auto-augmentation strategy to generate contrastive samples while preserving the most representative structures of the original graph. These graph-level representation learning methods are all based on the InfoMax principle [Linsker, 1988]. Note that there are many other graph representation learning methods such as VGAE [Kipf and Welling, 2016; Hamilton *et al.*, 2017; Cui *et al.*, 2020], graph embedding [Wu *et al.*, 2020; Yu *et al.*, 2021; Bai *et al.*, 2019; Verma and Zhang, 2019], self-supervised learning [Liu *et al.*, 2022; Hou *et al.*, 2022; Lee *et al.*, 2022; Xie *et al.*, 2022; Wu *et al.*, 2021; Rong *et al.*, 2020; Zhang *et al.*, 2021b,a; Xiao *et al.*, 2022], and contrastive learning [Le-Khac *et al.*, 2020; Qiu *et al.*, 2020; Ding *et al.*, 2022; Xia *et al.*, 2022; Fang *et al.*, 2022; Trivedi *et al.*, 2022; Han *et al.*, 2022; Mo *et al.*, 2022; Yin *et al.*, 2022; Xu *et al.*, 2021; Zhao *et al.*, 2021; Zeng and Xie, 2021; Li *et al.*, 2022a,b; Wei *et al.*, 2022], which will not be detailed in this paper due to the page length limit.

The InfoMax principle [Linsker, 1988], which is very popular in graph-level representation learning, advocates maximizing the mutual information between the representations of entire graphs and the representations of substructures of varying sizes [Peng *et al.*, 2020; Velickovic *et al.*, 2019; Hassani and Khasahmadi, 2020; Xie *et al.*, 2022; Qiu *et al.*, 2020]. These InfoMax-based methods usually evaluate the mutual information (MI) between different representations using Jensen-Shannon MI estimator [Sun *et al.*, 2019], following the formulations of $f$-GAN [Nowozin *et al.*, 2016] and Mutual Information Neural Estimation (MINE) [Belghazi *et al.*, 2018]. However, the Jensen-Shannon MI estimator necessitates the training of a neural network parameterized discriminator, which is overly complex. In addition, the estimator is based on sampling, which may not be accurate enough in exploiting the mutual information. As opposed to InfoMax, researchers proposed the graph information bottleneck (GIB) [Wu *et al.*, 2020] and the subgraph information bottleneck (SIB) [Yu *et al.*, 2021] that aim to learn the minimal sufficient representation for downstream tasks. But GIB [Wu *et al.*, 2020] and SIB [Yu *et al.*, 2021] may fail if the downstream tasks are not available in the representation learning stage.

In this work, we introduce a novel graph learning principle called Lovász principle, which is inspired by the Lovász number [Lovász, 1979] in graph theory. The Lovász number is an upper bound for a graph's Shannon capacity. It is closely associated with various global characteristics of a graph, such as the clique number and chromatic number of the complement graph. The handle vector for calculating the Lovász number is potentially a suitable choice for the graph-level representation, as it captures a graph's global features, though it suffers from a few difficulties. The contributions of this work are summarized as follows.

- We propose the Lovász principle, a novel framework for unsupervised graph representation learning. We show how to effectively and efficiently utilize the handle vectors to represent graphs. The Lovász principle exploits the topological structures of graphs globally via neural networks.

- We propose an enhanced Lovász principle via effectively incorporating subgraph Lovász numbers, while direct computation of subgraph Lovász numbers is extremely costly. The enhanced Lovász principle ensures similar graphs have similar representations.

- We extend the Lovász principles to semi-supervised representation learning. Note that it is possible to adapt the Lovász principles to more graph-based learning problems.

The experimental results of unsupervised learning, semi-supervised learning, and transfer learning on many benchmark graph datasets show that the proposed Lovász principles outperform graph kernels, classical graph embedding methods, and InfoMax principle based representation learning methods.

## 2 Notations and Preliminaries

In this work, we use $x$, $\boldsymbol{x}$, $\boldsymbol{X}$, $\mathcal{X}$ (or $X$) to denote scalar, vector, matrix, and set respectively. $\mathbf{1}_{a \times b}$ is a matrix of size $a \times b$ consisting only ones. $\boldsymbol{I}_n$ denotes an identity matrix of size $n \times n$. Let $G = (V, E)$ be a graph with $n$ nodes and $a$-dimensional node features $\{\boldsymbol{x}_v \in \mathbb{R}^a | v \in V\}$. We denote $\boldsymbol{A} \in \mathbb{R}^{n \times n}$ as the adjacency matrix and $\boldsymbol{X} = [\boldsymbol{x}_1, ..., \boldsymbol{x}_n]^\top \in \mathbb{R}^{n \times a}$ as the node features matrix. Let $\boldsymbol{z} \in \mathbb{R}^d$ be the $d$-dimensional graph-level representation of $G$, $\boldsymbol{h}_v \in \mathbb{R}^d$ be the $d$-dimensional node-level representation of node $v$, and $\boldsymbol{H} = [\boldsymbol{h}_1, ..., \boldsymbol{h}_n]^\top \in \mathbb{R}^{n \times d}$ be the node-level representations matrix of $G$. We denote $(p, q)$ as an edge between nodes $p, q$ and $(p, q) \in E$ if they are connected.

Let $\mathcal{G} = \{G_1, \ldots G_N\}$ be a dataset of $N$ graphs with $K$ classes, where $G_i = (V_i, E_i)$. For $G_i$, we denote its number of nodes as $n_i$, graph-level representation as $\boldsymbol{z}_i$, the adjacency matrix as $\boldsymbol{A}_i$, the node feature matrix as $\boldsymbol{X}_i$, and node-level representation matrix as $\boldsymbol{H}_i$. The graph-level representation matrix of dataset $\mathcal{G}$ is denoted as $\boldsymbol{Z} = [\boldsymbol{z}_1, ..., \boldsymbol{z}_N]^\top \in \mathbb{R}^{N \times d}$. The set of all node-level representations is denoted as $\mathcal{H} = \{\boldsymbol{H}_1, \ldots, \boldsymbol{H}_N\}$.

### 2.1 Lovász number

The definition of Lovász number [Lovász, 1979] is based on orthonormal representations of a graph. Therefore we first introduce the definition of orthonormal representations.

**Definition 2.1** (Orthonormal representations). Given a graph $G = (V, E)$ with $|V| = n$. Let

$$\mathcal{U} := \{\boldsymbol{U} \in \mathbb{R}^{d \times n} : \|\boldsymbol{u}_p\|_2 = 1, p = 1, 2, \ldots, n; \ \boldsymbol{u}_p^\top \boldsymbol{u}_q = 0, \ \forall (p, q) \notin E\}, \tag{1}$$

where $\boldsymbol{u}_p$ is the $p$-th column of $\boldsymbol{U}$. Then every $\boldsymbol{U} \in \mathcal{U}$ is an orthonormal representation of $G$ in $\mathbb{R}^d$.

Clearly, every graph has at least one orthonormal representation. For example, a trivial representation is that each node $p$ is represented by the standard basis vector $\boldsymbol{e}_p$. Based on Definition 2.1, we introduce the Lovász number [Lovász, 1979] of a graph as follows.

**Definition 2.2** (Lovász number). The Lovász number of a graph $G = (V, E)$ is defined as

$$\vartheta(G) := \min_{\boldsymbol{c}, \boldsymbol{U} \in \mathcal{U}} \max_{p \in V} \frac{1}{(\boldsymbol{c}^\top \boldsymbol{u}_p)^2}, \tag{2}$$

where $\boldsymbol{c} \in \mathbb{R}^d$ ranges over all unit vectors. The vector $\boldsymbol{c}$ yielding the minimum for (2), denoted by $\boldsymbol{c}^*$, is called the *handle* of the representation, where the corresponding $\boldsymbol{U}$ is denoted as $\boldsymbol{U}^*$ for convenience. $\boldsymbol{U}^*$ is called the optimal representation of $G$ in $\mathbb{R}^d$.

László Lovász provided a pentagon example, shown in Figure 1, to explain Lovász number defined by (2). The visualization of $\boldsymbol{U}^*$ and $\boldsymbol{c}^*$ of a pentagon is like an umbrella whose handle is $\boldsymbol{c}^*$ and the ribs are the five columns of $\boldsymbol{U}^*$. These five disjoint node pairs, i.e., $(\boldsymbol{u}_1^*, \boldsymbol{u}_3^*), (\boldsymbol{u}_1^*, \boldsymbol{u}_4^*), (\boldsymbol{u}_2^*, \boldsymbol{u}_4^*), (\boldsymbol{u}_2^*, \boldsymbol{u}_5^*), (\boldsymbol{u}_3^*, \boldsymbol{u}_5^*)$, are orthogonal to each other in visualization.

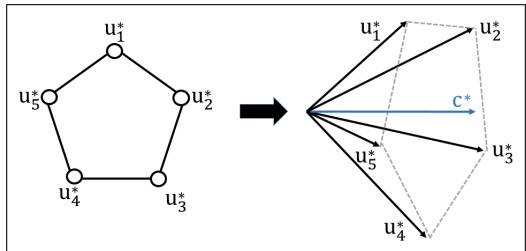

Figure 1: Pentagon example for Lovász number

The Lovász number $\vartheta(G)$ is an upper bound on the Shannon capacity of a graph $G$ and is polynomial-time computable [Grötschel *et al.*, 1981; Galli and Letchford, 2017] (e.g., using semidefinite programming (SDP)). Let $\bar{G}$ be the complement of $G$ with its clique number $\omega(\bar{G})$ and chromatic number $\chi(\bar{G})$. The Lovász "sandwich theorem" shows that the Lovász number is bounded between the clique number and the chromatic number of $\bar{G}$, i.e., $\omega(\bar{G}) \leq \vartheta(G) \leq \chi(\bar{G})$. Although computing $\omega(\bar{G})$ and $\chi(\bar{G})$ are both NP-hard, the "sandwich theorem" allows us to estimate the clique number and the chromatic number of $\bar{G}$ using only $\vartheta(G)$. More detailed discussion can be found in [Knuth, 1993].

### 2.2 Lovász theta kernel

Johansson *et al.* [2014] defined the Lovász theta kernel to evaluate the similarity between graphs. Suppose $S \subseteq V$ is a subset of the vertices of graph $G$, then the Lovász number of the subgraph

induced by $S$ is defined as

$$\vartheta_S(G) := \min_{\boldsymbol{c}} \max_{p \in S} \frac{1}{(\boldsymbol{c}^\top \boldsymbol{u}_p)^2}, \tag{3}$$

where $\boldsymbol{U}$ was pre-computed by Eq. (2) and $\boldsymbol{c}$ ranges over all unit vectors.

**Definition 2.3** (Lovász-$\vartheta$ kernel [Johansson *et al.*, 2014]). Let $k$ be a base kernel. The Lovász theta kernel between graphs $G = (V, E)$ and $G' = (V', E')$ is defined as

$$k_{\text{Lo}}(G, G') = \sum_{S \subseteq V} \sum_{S' \subseteq V'} \frac{\delta(|S|, |S'|)}{C_{S,S'}} k(\vartheta_S(G), \vartheta_{S'}(G')), \tag{4}$$

where $C_{S,S'} = \binom{|V|}{|S|}\binom{|V'|}{|S'|}$, $\delta(|S|, |S'|) = 1$ if $|S| = |S'|$, and $\delta(|S|, |S'|) = 0$ otherwise.

$k_{\text{Lo}}$ is a positive semi-definite kernel [Johansson *et al.*, 2014]. It is able to capture global properties of graphs and has been shown useful in SVM-based graph classification [Johansson *et al.*, 2014].

## 3  Lovász Principle for Graph Representation Learning

The Lovász number $\vartheta(G)$ of a graph $G$ provides an insight into the global property of the graph. It is a unique and deterministic value associated with an orthonormal representation $\boldsymbol{U}^*$ and a unit *handle* vector $\boldsymbol{c}^*$. The umbrella example in Figure 1 explains how to compute the Lovász number: compacting the ribs (i.e. $\boldsymbol{U}^*$) as much as possible and using $\boldsymbol{c}^*$ as the handle. This example provides intuition that the handle vector $\boldsymbol{c}^*$ is a natural and suitable representation of the graph $G$.

Given $\mathcal{G} = \{G_1, G_2, \ldots, G_N\}$ drawn from an unknown distribution $\mathcal{D}_G$, we want to represent each graph as a vector such that these vectors preserve some important information of $\mathcal{D}_G$. Suppose we have an algorithm $\mathcal{A}$ such that

$$(\boldsymbol{U}_i^*, \boldsymbol{c}_i^*) = \mathcal{A}(G_i), \quad i = 1, 2, \ldots, N, \tag{5}$$

where $\mathcal{A}$ is some solver for (2). It is natural to use $\boldsymbol{c}_1^*, \boldsymbol{c}_2^*, \ldots, \boldsymbol{c}_N^*$ as representations of $G_1, G_2, \ldots, G_N$ respectively. However, this method has the following limitations[‡].

    i. **Non-uniqueness** For any $G_i$, both $\boldsymbol{U}_i^*$ and $\boldsymbol{c}_i^*$ are not unique. For example, let $\boldsymbol{Q} \in \mathbb{R}^{d \times d}$ be an orthonormal matrix, i.e., $\boldsymbol{Q}^\top \boldsymbol{Q} = \boldsymbol{Q}\boldsymbol{Q}^\top = \boldsymbol{I}_d$, and let $\boldsymbol{c}_i' = \boldsymbol{Q}\boldsymbol{c}_i^*$ and $\boldsymbol{U}_i' = \boldsymbol{Q}\boldsymbol{U}_i^*$. We have $\|\boldsymbol{c}'\|_2 = 1$, $(\boldsymbol{U}')^\top \boldsymbol{U} = (\boldsymbol{U}_i^*)^\top \boldsymbol{U}_i^*$, and $(\boldsymbol{c}')^\top \boldsymbol{u}_p' = \vartheta(G_i)$. This means $(\boldsymbol{c}_i^*, \boldsymbol{U}_i^*)$ and $(\boldsymbol{c}_i', \boldsymbol{U}')$ yield the same Lovász number for $G_i$, though they could be very different. Thus, for two graphs $G_i$ and $G_j$ in $\mathcal{G}$, even when they are isomorphic, $\boldsymbol{c}_i^*$ and $\boldsymbol{c}_j^*$ could be very different. However, for graph representation, we hope that similar graphs have similar representations. For two graphs $G_i$ and $G_j$, one may align their orthonormal representations using $\hat{\boldsymbol{Q}} = \operatorname{argmin}_{\boldsymbol{Q}^\top \boldsymbol{Q} = \boldsymbol{I}_d} \|\boldsymbol{U}_i^* - \boldsymbol{Q}\boldsymbol{U}_j^*\|_F^2$ and compare them according to $\|\boldsymbol{c}_i^* - \boldsymbol{c}_j^*\hat{\boldsymbol{Q}}\|_2$. This however only works when the $n_i = n_j$ and $G_i$ and $G_j$ are matched.

    ii. **High computational cost** For each $G_i$ in $\mathcal{G}$, we need to solve the optimization problem (2), for which the time complexity of SDP is at least $\mathcal{O}(|E_i| n_i^{2.5})$ [Jiang *et al.*, 2020]. Thus the total time complexity for $\mathcal{G}$ is $\mathcal{O}(\sum_{i=1}^N |E_i| n_i^{2.5})$. Therefore, this representation method is not scalable to large datasets.

    iii. **Ignorance of node features** The computation of (5) solely relies on the graph structure and does not take advantage of the node feature matrix $\boldsymbol{X}_i$ that is often available and informative.

    iv. **Non-generalization** Suppose we have some new graphs and want to obtain their representations. We cannot utilize the representations of $\mathcal{G}$ and we have to solve (2) again for each new graph.

    v. **Non-global sensing** The computation of (5) treats each graph separately and cannot effectively take advantage of the global information or structure of $\mathcal{G}$. Individual graphs may have noise or outliers, which cannot be handled by a local method.

---

[‡]In our experiments (Table 1), this naive method, termed as LovászNum, is tested. In addition, the Lovász-$\vartheta$ kernel introduced in Section 2.2 is also tested.

To solve the aforementioned five issues, we present a machine learning method. We use a neural network $\mathcal{F}_W$ (parameterized by $W$) to approximate $\mathcal{A}$. $\mathcal{F}_W$ can be learned from $\mathcal{G}$ as well as some additional information such as the node feature matrices $\{\boldsymbol{X}_1, \ldots, \boldsymbol{X}_N\}$. Specifically, we hope that

$$(\boldsymbol{U}_i^*, \boldsymbol{c}_i^*) \approx \mathcal{F}_W(\boldsymbol{A}_i, \boldsymbol{X}_i), \quad i = 1, 2, \ldots, N. \tag{6}$$

Thus, $\mathcal{F}_W$ plays a role representing a graph (drawn from $\mathcal{D}_G$) to a matrix of nodes representation and a vector of graph representation. For new graphs sampled from $\mathcal{D}_G$, $\mathcal{F}_W$ should generalize well when the approximation errors in (6) are small enough and $\mathcal{F}_W$ is not too complex. For convenience, we split $\mathcal{F}_W$ into two parts, i.e., $\mathcal{F}_W(\cdot, \cdot) = (F(\cdot, \cdot; \theta), f(\cdot, \cdot; \phi))$, though $F$ and $f$ can share some parameters. We let $\boldsymbol{U}_i^* \approx F(\boldsymbol{A}_i, \boldsymbol{X}_i; \theta)$ and $\boldsymbol{c}_i^* \approx f(\boldsymbol{A}_i, \boldsymbol{X}_i; \phi)$. $F(\cdot, \cdot; \theta)$ is the model of node-level representation learning while $f(\cdot, \cdot; \phi)$ is the model of graph-level representation learning. We let

$$\boldsymbol{H}_i^\theta := F(\boldsymbol{A}_i, \boldsymbol{X}_i; \theta), \text{ and } \boldsymbol{z}_i^\phi := f(\boldsymbol{A}_i, \boldsymbol{X}_i; \phi), \ \forall i = 1, 2, ..., N. \tag{7}$$

We denote the graph-level representations matrix as $\boldsymbol{Z}_\phi = [\boldsymbol{z}_1^\phi, ..., \boldsymbol{z}_N^\phi]^\top$ and the node-level representations set as $\mathcal{H}_\theta = \{\boldsymbol{H}_1^\theta, ..., \boldsymbol{H}_N^\theta\}$. To achieve (6), we propose to solve

$$\underset{\phi, \theta}{\text{minimize}} \sum_{i=1}^N \left\{ \underbrace{\max_{p \in V_i} \frac{1}{\left((\boldsymbol{z}_i^\phi)^\top \boldsymbol{h}_p^\theta\right)^2}}_{\ell_1} + \mu \bigg( \underbrace{\left\| \boldsymbol{M}_i \odot \left( \boldsymbol{H}_i^\theta (\boldsymbol{H}_i^\theta)^\top - \boldsymbol{I}_{n_i} \right) \right\|_F^2}_{\ell_2} + \underbrace{\left( (\boldsymbol{z}_i^\phi)^\top \boldsymbol{z}_i^\phi - 1 \right)^2}_{\ell_3} \bigg) \right\},$$
$$\tag{8}$$

where $\boldsymbol{M}_i = \boldsymbol{1}_{n_i \times n_i} - \boldsymbol{A}_i$ is a mask matrix and $\mu > 0$ is a regularization parameter. The roles of $\ell_1$, $\ell_2$, and $\ell_3$ in (8) are explained as follows.

- $\ell_1$ corresponds to the objective in the definition of Lovász number of $G_i$.
- $\ell_2$ is to approximate the orthonormal representation for $G_i$, i.e., $(\boldsymbol{h}_p^\theta)^\top \boldsymbol{h}_q^\theta \approx 0$ if $(p, q) \notin E_i$ and $\|\boldsymbol{h}_p\|_2 \approx 1 \ \forall p \in V_i$.
- $\ell_3$ corresponds to the unit-length requirement for the handle vector of $G_i$, i.e., $\|\boldsymbol{z}_i^\phi\|_2 \approx 1$.

We call (8) **Lovász principle**[§], since it aims to learn an $\mathcal{F}_W$ to solve the optimization of Lovász number for the graphs drawn from $\mathcal{D}_G$. It is known that the Lovász number $\vartheta(G)$ is an upper bound on the Shannon capacity of $G = (V, E)$. The Shannon capacity [Shannon, 1956] models the amount of information that can be transmitted across a noisy communication channel, where certain signal values can be confused with each other. Here, one signal value corresponds to one node of $G$ and $(p, q) \in E$ means that the corresponding two signals can be confused with each other. Therefore, the graph-level and node-level representations given by our Lovász principle correspond to the upper bound of the amount of information transmitted over the graph that is distinguishable between nodes.

Note that instead of the regularized unconstrained optimization (8), we can also use constrained optimization ($\ell_2 = \ell_3 = 0$), which we call strict Lovász principle. We may use the Lagrange multipliers method, projected gradient descent, or exact (or inexact) penalty method to solve the constrained optimization. Take the inexact penalty method as an example, we just need to increase the $\mu$ in (8) gradually in the optimization. The graph representation performance comparison between unconstrained and constrained optimizations will be shown in Section 6.5 and Appendix E.

For convenience, we let

$$\mathcal{L}_{\text{Lo}} := \sum_{i=1}^{|\mathcal{G}|} \max_{p \in V_i} \frac{1}{((\boldsymbol{z}_i^\phi)^\top \boldsymbol{h}_p^\theta)^2} + \mu \left( \left\| \boldsymbol{M}_i \odot \left( \boldsymbol{H}_i^\theta (\boldsymbol{H}_i^\theta)^\top - \boldsymbol{I}_{n_i} \right) \right\|_F^2 + \left( (\boldsymbol{z}_i^\phi)^\top \boldsymbol{z}_i^\phi - 1 \right)^2 \right), \tag{9}$$

and call it Lovász loss. The Lovász loss is mainly designed for unsupervised graph-level representation learning [Wu *et al.*, 2022; Maron *et al.*, 2019; Oono and Suzuki, 2019; Ståhlberg *et al.*, 2022], which can be used as an alternative to the popular InfoMax loss [Linsker, 1988] (see (16)).

**Lovász principle for semi-supervised learning**    Inspired by InfoGraph [Sun *et al.*, 2019] (see (17)), we propose a Lovász loss function for semi-supervised learning tasks. Suppose the dataset $\mathcal{G}$ has

---

[§]We also provide an equivalent formulation based on the complement graph of $G$ in Appendix A.

two subsets: a labeled dataset $\mathcal{G}^L$ and an unlabeled dataset $\mathcal{G}^U$. Then we deploy another supervised encoder with parameter $\psi$ and generate the supervised node-level representations $\boldsymbol{H}_i^{\psi}$, graph-level representations $\boldsymbol{z}_i^{\psi}$, and then prediction $\hat{\boldsymbol{y}}_i^{\psi}$. The overall loss function is

$$\mathcal{L}_{\text{Lo-semi}} := \sum_{l=1}^{|\mathcal{G}^L|} \ell_{\text{supervised}}(\hat{\boldsymbol{y}}_l^{\psi}, \boldsymbol{y}_l) + \mathcal{L}_{\text{unsupervised}}(\mathcal{G}) + \lambda \sum_{i=1}^{|\mathcal{G}|} \left\| \boldsymbol{z}_i^{\phi} - \boldsymbol{z}_i^{\psi} \right\|_2^2, \tag{10}$$

where $\lambda$ is a positive hyperparameter, the supervised loss $\ell_{\text{supervised}}$ is the cross-entropy loss, and the unsupervised loss $\mathcal{L}_{\text{unsupervised}}$ is the Lovász loss $\mathcal{L}_{\text{Lo}}$ (Eq. (9)) or the enhanced Lovász loss $\mathcal{L}_{\text{ELo}}$ (Eq. (14)). The last term encourages the representations learned by the two encoders to be similar.

## 4 Enhancing Lovász Principle with Subgraph Lovász Number

Lovász principle does not explicitly utilize the Lovász number in graph embedding, though the Lovász numbers of subgraphs can be useful in comparing graphs [Johansson *et al.*, 2014]. Therefore, we propose to use subgraph Lovász number to enhance Lovász principle based graph representation learning. We may consider taking advantage of the Lovász-$\vartheta$ kernel proposed by [Johansson *et al.*, 2014]. However, we encounter the following two difficulties.

  i. Computing the Lovász numbers (3) of subgraphs is time-consuming because we need to solve (2) for every graph and the number of subgraphs of each graph is often very large (up to $2^{|V|}$). Hence, for large graph dataset, we cannot use (4) directly.

  ii. The Lovász-$\vartheta$ kernel (4) is a pair-wise method and cannot effectively exploit the global structure of $\mathcal{G}$.

To solve the aforementioned problems, we present an iterative-refinement strategy that computes the subgraph Lovász numbers using the embeddings given by the Lovász principle. Specifically, at iteration $t$, we have the graph-level representations $\boldsymbol{Z}_{\phi}^{(t-1)}$ and the node-level representations $\mathcal{H}_{\theta}^{(t-1)}$ given by iteration $t-1$. Inspired by the Lovász-$\vartheta$ kernel (4), we compute the similarity between graph $G_i$ and $G_j$ as

$$K_{ij}^{(t-1)} = \sum_{S_i \subseteq V_i} \sum_{S_j \subseteq V_j} \frac{\delta(|S_i|, |S_j|)}{C_{S_i, S_j}} k(\vartheta_{S_i}^{(t-1)}(G_i), \vartheta_{S_j}^{(t-1)}(G_j)), \tag{11}$$

where $C_{S_i, S_j} = \binom{|V_i|}{|S_i|} \binom{|V_j|}{|S_j|}$ and $\vartheta_{S_i}^{(t-1)}(G_i)$ (similar for $G_j$) is obtained by

$$\vartheta_{S_i}^{(t-1)}(G_i) = \max_{p \in S_i} \frac{1}{(\boldsymbol{z}_i^{(t-1)\top} \boldsymbol{h}_p^{(t-1)})^2}. \tag{12}$$

The computation of $1/(\boldsymbol{z}_i^{(t-1)\top} \boldsymbol{h}_p^{(t-1)})^2$ for every $p \in V_i$ was already done when computing $\mathcal{L}_{\text{Lo}}$ via (9) at iteration $t-1$ and there is no need to solve (3). For (11), we do not need to consider all possible subgraphs and we can just randomly sample subgraphs with some fixed sizes (numbers of nodes), which is similar to the truncated Lovász-$\vartheta$ kernel of [Johansson *et al.*, 2014]. Thus we can obtain the similarity $K_{ij}^{(t-1)}$ efficiently. Adapting the idea of spectral embedding [Belkin and Niyogi, 2001], we propose the following subgraph Lovász number (SLN) loss (at iteration $t$)

$$\mathcal{L}_{\text{SLN}}^{(t)} := \sum_{i=1}^{|\mathcal{G}|} \sum_{j=1}^{|\mathcal{G}|} K_{ij}^{(t-1)} \left\| \boldsymbol{z}_i^{\phi} - \boldsymbol{z}_j^{\phi} \right\|_2^2 + \gamma \left( \left\| \boldsymbol{Z}_{\phi}^{\top} \boldsymbol{Z}_{\phi} - \boldsymbol{I}_d \right\|_F^2 + \left\| \boldsymbol{Z}_{\phi}^{\top} \boldsymbol{1}_{N \times 1} \right\|_2^2 \right), \tag{13}$$

where $\gamma > 0$. The two regularization terms in $\mathcal{L}_{\text{SLN}}^{(t)}$ aim to make the graph-level representations orthonormal and centered, which is consistent with the constraints in spectral embedding. Minimizing $\mathcal{L}_{\text{SLN}}^{(t)}$ encourages that the graph-level representations of similar graphs (in the sense of subgraph Lovász numbers) are closer to each other at iteration $t$. Integrating (13) with (9), we obtain the following enhanced Lovász loss at iteration $t$

$$\mathcal{L}_{\text{ELo}}^{(t)} := \mathcal{L}_{\text{Lo}}^{(t)} + \eta \mathcal{L}_{\text{SLN}}^{(t)}, \tag{14}$$

where $\eta > 0$ is a hyperparameter. It is worth noting that $\mathcal{L}_{\text{ELo}}$ as well as $\mathcal{L}_{\text{Lo}}$ can be implemented batch-wisely, via replacing $\mathcal{G}$ with its subsets. Similar to $\mathcal{L}_{\text{Lo}}$, $\mathcal{L}_{\text{ELo}}$ can also be applied to semi-supervised graph classification, i.e., (10).

# 5 Related Work

Besides the Lovász-$\vartheta$ introduced in Section 2.2, the closest work to our Lovász principle is the InfoMax principle. Following [Nowozin *et al.*, 2016; Sun *et al.*, 2019; Belghazi *et al.*, 2018], suppose the node-level representation $\boldsymbol{h}_p(x)$ and the graph-level representation $\boldsymbol{z}(x)$ are depending on the input $x$, $T_\varphi$ is a discriminator parameterized by a neural network with parameters $\varphi$, the Jensen-Shannon mutual information (MI) estimator [Fuglede and Topsoe, 2004; Nowozin *et al.*, 2016; Hjelm *et al.*, 2019; Sun *et al.*, 2019] $I_\varphi$ between $\boldsymbol{h}_p$ and $\boldsymbol{z}$ is defined as

$$I_\varphi(\boldsymbol{h}_p, \boldsymbol{z}) = \mathbb{E}_{\mathbb{P}}[-\mathrm{sp}(-T_\varphi(\boldsymbol{h}_p(x), \boldsymbol{z}(x)))] - \mathbb{E}_{\mathbb{P} \times \tilde{\mathbb{P}}}[\mathrm{sp}(T_\varphi(\boldsymbol{h}_p(x'), \boldsymbol{z}(x)))], \tag{15}$$

where $x$ is the input sample from distribution $\mathbb{P}$, $x'$ is the negative sample from distribution $\tilde{\mathbb{P}}$, and $\mathrm{sp}(a) = \log(1 + e^a)$ denotes the softplus function. Many recent graph-level representation learning methods [Sun *et al.*, 2019; You *et al.*, 2020; Yin *et al.*, 2022] are based on the InfoMax principle, i.e., maximizing (15). For instance, the InfoGraph proposed by [Sun *et al.*, 2019] obtains graph-level representations by maximizing the mutual information between the graph-level representation and the node-level representations as follows

$$\phi^*, \theta^*, \varphi^* = \arg\max_{\phi, \theta, \varphi} \sum_{i=1}^{|\mathcal{G}|} \frac{1}{|V_i|} \sum_{p \in V_i} I_\varphi(\boldsymbol{h}_p^\theta, \boldsymbol{z}_i^\phi) \triangleq -\mathcal{L}_{\text{unsupervised}}^{I_\varphi}(\mathcal{G}). \tag{16}$$

For semi-supervised learning, the dataset $\mathcal{G}$ is split into labeled dataset $\mathcal{G}^L$ and unlabeled dataset $\mathcal{G}^U$. They deploy another supervised encoder with parameter $\psi$ and then generate the supervised node-level representations $\boldsymbol{H}_i^\psi$, graph-level representations $\boldsymbol{z}_i^\psi$ and prediction $\hat{\boldsymbol{y}}_i^\psi$. The loss function of InfoGraph for semi-supervised learning is defined as follows

$$\mathcal{L}_{\text{info-semi}} = \sum_{l=1}^{|\mathcal{G}^L|} \ell_{\text{supervised}}(\hat{\boldsymbol{y}}_l^\psi, \boldsymbol{y}_l) + \mathcal{L}_{\text{unsupervised}}^{I_\varphi}(\mathcal{G}) - \lambda \sum_{i=1}^{|\mathcal{G}|} \frac{1}{|V_i|} I_\varphi(\boldsymbol{z}_i^\phi . \boldsymbol{z}_i^\psi). \tag{17}$$

The comparison between the InfoMax principle and our Lovász principle is as follows.

- The InfoMax principle focuses on the mutual information between graph-level representation and node-level representation, while our Lovász principle is derived from the Lovász number, a fundamental topological property of graph.

- Our Lovász principle only needs to optimize $\phi$ and $\theta$. Differently, besides $\phi$ and $\theta$, the InfoMax principle has to optimize an additional discriminator parameter $\varphi$ for the Jensen-Shannon MI estimator. Thus, our Lovász principle is simpler than the InfoMax principle.

- Approximating mutual information using neural network is challenging [Nowozin *et al.*, 2016] and the Jensen-Shannon MI estimator $I_\varphi$ only provides an approximation by sampling rather than an exact computation. In contrast, our Lovász principle does not rely on mutual information and sampling.

It is worth noting that the Lovász convolutional networks (LCN) proposed by [Yadav *et al.*, 2019] was motivated by the observation that removing certain vertices from a graph doesn't affect the graph's global properties such as the Lovász number. LCN does not involve any optimization related to the Lovász number and was designed as an alternative to GCN. Our Lovász principle is an optimization principle that can be used in any graph neural network (e.g. LCN). It is also useful in many applications such as graph prompt learning [Liu *et al.*, 2023; Sun *et al.*, 2022] and graph anomaly detection [Ma *et al.*, 2021; Zhang *et al.*, 2023; Cai *et al.*, 2023].

# 6 Experiments

In this section, we evaluate the effectiveness of the Lovász principle compared to the InfoMax principle in graph representation learning methods and a few other baselines such as graph kernels. The graph representation learning methods we considered in this paper include InfoGraph [Sun *et al.*, 2019], GraphCL [You *et al.*, 2020], AD-GCL [Suresh *et al.*, 2021], JOAO [You *et al.*, 2021], and AutoGCL [Yin *et al.*, 2022], which are the most current and influential methods spanning from

2019 to 2022. Note that the graph-level representation learning principles (GIB) [Wu *et al.*, 2020] and (SIB) [Yu *et al.*, 2021] are not suitable for unsupervised graph learning and hence will not be compared in this work. To ensure fair comparisons, we follow the neural network architectures of those InfoMax based methods and replace the InfoMax loss (Eq. (16)) with the Lovász loss $\mathcal{L}_{Lo}$ (Eq. (9)) or enhanced Lovász loss $\mathcal{L}_{ELo}$ (Eq. (14)) while keeping the network structures and parameter settings unchanged. We conduct the experiments on TUD benchmark datasets [Morris *et al.*, 2020] and ChEMBL benchmark datasets [Mayr *et al.*, 2018; Gaulton *et al.*, 2012].

## 6.1 Unsupervised Learning

The first category of compare methods is graph kernel-based methods such as graphlet kernel (GL) [Shervashidze *et al.*, 2009], Weisfeiler-Lehman sub-tree kernel (WL) [Shervashidze *et al.*, 2011], deep graph kernel (DGK) [Yanardag and Vishwanathan, 2015], and Lovász-$\vartheta$ kernel [Johansson *et al.*, 2014]. The second category is traditional unsupervised graph representation methods like node2vec [Grover and Leskovec, 2016], sub2vec Adhikari *et al.* [2018], and graph2vec [Narayanan *et al.*, 2017]. The third category is the methods based on the InfoMax principle (Eq. (16)), including InfoGraph [Sun *et al.*, 2019], GraphCL [You *et al.*, 2020], AD-GCL [Suresh *et al.*, 2021], JOAOv2 [You *et al.*, 2021], and AutoGCL [Yin *et al.*, 2022]. We also include the Lovász number method (LovászNum) [Lovász, 1979] as a baseline, where we solve the Lovász number (Eq. (2)) using semidefinite programming (SDP) [Wolkowicz *et al.*, 2012] and then use the handle vector $c^*$ as the graph-level representation.

Following [Sun *et al.*, 2019; You *et al.*, 2021; Yin *et al.*, 2022], we train a graph representation model on unlabeled data to obtain graph representations and use these representations and graph labels to train a classifier. Our experimental setup is similar to that of AutoGCL [Yin *et al.*, 2022]. Specifically, we use a 5-layer GIN [Xu *et al.*, 2018] with hidden size 128 as the representation model and an SVM as the classifier. The model is trained with a batch size of 128 and a learning rate of 0.001. For those contrastive learning methods (e.g., JOJOv2 and AutoGCL), we use 30 epochs of contrastive pre-training under the naive strategy. We perform 10-fold cross-validation on each dataset and repeat 10 times with different random seeds and record the average accuracy (ACC) and standard deviation.

Table 1: Performance (ACC) of unsupervised learning. The baseline results are from AutoGCL [Yin *et al.*, 2022] and JOAO [You *et al.*, 2021]. The **bold**, blue and green numbers denote the best, second best and third best performances respectively, which also applies to Tables 2 and 3.

| | methods | MUTAG | PROTEINS | DD | NCI1 | COLLAB | IMDB-B | REDDIT-B | REDDIT-M5K |
|---|---|---|---|---|---|---|---|---|---|
| kernels | GL | 81.66±2.11 | - | - | - | - | 65.87±0.98 | 77.34±0.18 | 41.01±0.17 |
| | WL | 80.72±3.00 | 72.92±0.56 | - | 80.01±0.50 | - | 72.30±3.44 | 68.82±0.41 | 46.06±0.21 |
| | DGK | 87.44±2.72 | 73.30±0.82 | - | 80.31±0.46 | - | 66.96±0.56 | 78.04±0.39 | 41.27±0.18 |
| | Lovász-$\vartheta$ | 82.57±1.68 | 71.86±1.41 | - | 75.90±1.33 | - | 67.26±1.85 | 76.03±1.87 | 43.57±1.79 |
| vector embedding | node2vec | 72.63±10.20 | 57.49±3.57 | - | 54.89±1.61 | - | - | - | - |
| | sub2vec | 61.05±15.80 | 53.03±5.55 | - | 52.84±1.47 | - | 55.26±1.54 | 71.48±0.41 | 36.68±0.42 |
| | graph2vec | 83.15±9.25 | 73.30±2.05 | - | 73.22±1.81 | - | 71.10±0.54 | 75.78±1.03 | 47.86±0.26 |
| InfoMax principle | InfoGraph | 89.01±1.13 | 74.44±0.31 | 72.85±1.78 | 76.20±1.06 | 70.65±1.13 | 73.03±0.87 | 82.50±1.42 | 53.46±1.03 |
| | GraphCL | 86.80±1.34 | 74.39±0.45 | 78.62±0.40 | 77.87±0.41 | 71.36±1.15 | 71.14±0.44 | 89.53±0.84 | 55.99±0.28 |
| | AD-GCL | 87.13±1.56 | 73.59±0.65 | 74.49±0.52 | 69.67±0.51 | 73.32±0.61 | 71.57±1.01 | 85.52±0.79 | 53.00±0.82 |
| | JOAOv2 | 86.91±1.01 | 71.25±0.85 | 66.91±1.75 | 72.99±0.75 | 70.40±2.21 | 71.60±0.86 | 78.35±1.38 | 55.57±2.86 |
| | AutoGCL | 88.64±1.08 | 75.80±0.36 | 77.57±0.60 | 82.00±0.29 | 70.12±0.68 | 73.30±0.40 | 88.58±1.49 | 56.75±0.18 |
| | LovászNum | 81.24±1.59 | 62.46±1.31 | 67.65±2.31 | 74.73±1.88 | 72.47±1.83 | 70.57±1.73 | 71.25±1.59 | 43.24±1.72 |
| Lovász principle (use $\mathcal{L}_{Lo}$) | InfoGraph | 89.67±1.54 | 75.26±1.43 | 74.13±1.49 | 78.21±1.35 | 71.46±1.21 | 73.87±1.32 | 84.76±1.86 | 54.57±1.38 |
| | GraphCL | 87.24±1.96 | 75.87±2.17 | 79.14±1.67 | 79.13±1.27 | 72.52±1.37 | 72.44±1.46 | 89.87±2.13 | 56.12±1.73 |
| | AD-GCL | 87.44±2.13 | 74.29±2.80 | 76.25±1.48 | 75.12±2.13 | 73.85±1.05 | 73.02±1.35 | 87.11±1.95 | 54.61±2.35 |
| | JOAOv2 | 87.19±1.92 | 73.15±1.46 | 73.15±2.17 | 74.15±1.67 | 72.62±1.43 | 72.18±1.72 | 84.19±1.67 | 53.74±1.70 |
| | AutoGCL | 89.02±1.47 | 76.23±1.25 | 78.95±1.39 | 82.63±2.12 | 71.31±1.72 | 73.95±1.36 | 89.41±1.81 | 57.28±1.62 |
| Lovász principle (use $\mathcal{L}_{ELo}$) | InfoGraph | 90.13±2.05 | 76.12±1.72 | 75.76±1.64 | 79.36±1.57 | 72.67±1.95 | 74.96±1.49 | 84.53±1.79 | 55.12±1.47 |
| | GraphCL | 87.93±2.42 | 76.82±1.34 | 77.35±1.95 | 80.11±1.47 | 74.16±1.37 | 73.87±1.52 | 90.23±1.87 | 56.83±1.35 |
| | AD-GCL | 88.50±1.82 | 75.22±1.93 | 76.14±1.21 | 78.15±1.81 | 74.57±1.98 | 73.48±1.41 | 88.16±1.37 | 55.64±1.63 |
| | JOAOv2 | 88.76±1.43 | 75.27±1.61 | 74.62±2.58 | 76.23±1.75 | 72.85±1.73 | 72.97±1.37 | 85.31±1.48 | 54.68±1.48 |
| | AutoGCL | 89.87±1.85 | 76.03±1.37 | 79.31±1.27 | 82.95±1.26 | 72.23±1.52 | 74.52±1.44 | 90.65±1.46 | 57.93±1.72 |

As shown in Table 1, the enhanced Lovász loss $\mathcal{L}_{ELo}$ (Eq. (14)) achieves the best performance on all datasets. By replacing the InfoMax loss with the Lovász loss $\mathcal{L}_{Lo}$, the performances of the five graph representation learning methods are improved, which demonstrates the effectiveness of the Lovász principle. Furthermore, $\mathcal{L}_{ELo}$ outperformed $\mathcal{L}_{Lo}$ in most cases, which verified the effectiveness of introducing subgraph Lovász numbers to the Lovász principle. It is worth noting that the LovászNum method performs worse than the Lovász principle based methods, which confirms the limitations we analyzed in Section 3 and verifies the necessity and significance of our proposed methods.

## 6.2 Semi-supervised Learning

Following [Hu *et al.*, 2019; You *et al.*, 2021; Yin *et al.*, 2022], we compare Lovász principle with InfoMax principle in semi-supervised learning tasks. The semi-supervised losses of our Lovász principle based methods and InfoMax based methods $\mathcal{L}_{\text{info-semi}}$ were shown in (10) and (17) respectively. Following the settings of AutoGCL [Yin *et al.*, 2022], we employ a 10-fold cross-validation on each dataset. For each fold, we use 80% of the total data as the unlabeled data, 10% as labeled training data, and 10% as labeled testing data. The classifier for labeled data is a ResGCN [Chen *et al.*, 2019] with 5 layers and a hidden size of 128. We repeat each experiment 10 times and report the average accuracy in Table 2. We see that our Lovász loss $\mathcal{L}_{\text{Lo}}$ and the enhanced Lovász loss $\mathcal{L}_{\text{ELo}}$ outperformed InfoMax loss in all cases. Furthermore, $\mathcal{L}_{\text{ELo}}$ outperformed $\mathcal{L}_{\text{Lo}}$ in most cases. These results are consistent with those in Secion 6.1.

Table 2: Performance (ACC) of semi-supervised learning.

| | methods | NCI1 | PROTEINS | DD | COLLAB | REDDIT-B | REDDIT-M5K | GITHUB |
|---|---|---|---|---|---|---|---|---|
| | *no Pretrain* | 73.72±0.24 | 70.40±1.54 | 73.56±0.41 | 73.71±0.27 | 86.63±0.27 | 51.33±0.44 | 60.87±0.17 |
| Pretrain-GNN | Infomax | 74.86±0.26 | 72.27±0.40 | 75.78±0.34 | 73.76±0.29 | 88.66±0.95 | 53.61±0.31 | 65.21±0.88 |
| | ContextPred | 73.00±0.30 | 70.23±0.63 | 74.66±0.51 | 73.69±0.37 | 84.76±0.52 | 51.23±0.84 | 62.35±0.73 |
| InfoMax principle | GraphCL | 74.63±0.25 | 74.17±0.34 | 76.17±1.37 | 74.23±0.21 | 89.11±0.19 | 52.55±0.45 | 65.81±0.79 |
| | AD-GCL | 75.18±0.31 | 73.96±0.47 | 77.91±0.73 | 75.82±0.26 | 90.10±0.15 | 53.49±0.28 | 64.17±1.38 |
| | JOAOv2 | 74.86±0.39 | 73.31±0.48 | 75.81±0.73 | 75.53±0.18 | 88.79±0.65 | 52.71±0.28 | 66.66±0.60 |
| | AutoGCL | 73.75±2.25 | 75.65±2.40 | 77.50±4.41 | 77.16±1.48 | 79.80±3.47 | 49.91±2.70 | 62.46±1.51 |
| Lovász principle (use $\mathcal{L}_{\text{Lo}}$) | GraphCL | 75.46±1.53 | 75.12±1.87 | 77.46±1.52 | 76.12±1.15 | 89.87±1.68 | 53.69±1.68 | 66.72±1.53 |
| | AD-GCL | 76.62±1.83 | 74.21±1.71 | 78.27±1.39 | 76.27±1.74 | 90.36±1.56 | 54.06±1.32 | 65.32±1.04 |
| | JOAOv2 | 76.13±1.76 | 73.73±1.86 | 76.27±1.48 | 77.35±1.27 | 89.31±1.85 | 53.17±1.76 | 66.35±1.96 |
| | AutoGCL | 75.77±1.48 | 76.36±1.57 | 78.16±1.61 | 77.63±1.78 | 84.64±2.53 | 51.31±1.81 | 64.87±1.62 |
| Lovász principle (use $\mathcal{L}_{\text{ELo}}$) | GraphCL | 75.81±1.68 | 75.88±1.67 | 78.43±1.48 | 77.57±1.58 | 90.67±1.27 | 54.81±1.73 | 67.04±1.45 |
| | AD-GCL | 77.28±1.04 | 75.43±1.58 | 78.67±1.64 | 76.98±1.87 | 91.54±1.39 | 55.46±1.59 | 66.87±1.25 |
| | JOAOv2 | 76.25±1.59 | 74.67±1.37 | 77.96±1.86 | 78.84±1.75 | 90.25±1.22 | 54.32±1.89 | 67.52±1.73 |
| | AutoGCL | 76.53±1.92 | 76.89±1.55 | 78.82±1.90 | 78.46±1.39 | 87.31±1.57 | 53.17±1.50 | 66.47±1.26 |

## 6.3 Transfer Learning

Following [Hu *et al.*, 2019; You *et al.*, 2021; Yin *et al.*, 2022], we compare the performance of our Lovász principles with the InfoMax principle in the task of transfer learning. We use the Pretrain-GNN method [Hu *et al.*, 2019] as a baseline and employ the Infomax, EdgePred, AttrMasking, and ContextPred pre-training strategies. The experimental settings follow those of AutoGCL [Yin *et al.*, 2022]. More details are in the appendix. As shown in Table 3, the improved Lovász loss $\mathcal{L}_{\text{ELo}}$ performs the best on transfer learning tasks. In addition, the Lovász principle based methods generally outperform those based on the InfoMax principle in most cases.

Table 3: Performance (ROC-AUC score) of transfer learning.

| | methods | BBBP | Tox21 | ToxCast | SIDER | ClinTox | MUV | HIV | BACE |
|---|---|---|---|---|---|---|---|---|---|
| | *no Pretrain* | 65.8±4.5 | 74.0±0.8 | 63.4±0.6 | 57.3±1.6 | 58.0±4.4 | 71.8±2.5 | 75.3±1.9 | 70.1±5.4 |
| Pretrain-GNN's strategies | Infomax | 68.8±0.8 | 75.3±0.5 | 62.7±0.4 | 58.4±0.8 | 69.9±3.0 | 75.3±2.5 | 76.0±0.7 | 75.9±1.6 |
| | EdgePred | 67.3±2.4 | 76.0±0.6 | 64.1±0.6 | 60.4±0.7 | 64.1±3.7 | 74.1±2.1 | 76.3±1.0 | 79.9±0.9 |
| | AttrMasking | 64.3±2.8 | 76.7±0.4 | 64.2±0.5 | 61.0±0.7 | 71.8±4.1 | 74.7±1.4 | 77.2±1.1 | 79.3±1.6 |
| | ContextPred | 68.0±2.0 | 75.7±0.7 | 63.9±0.6 | 60.9±0.6 | 65.9±3.8 | 75.8±1.7 | 77.3±1.0 | 79.6±1.2 |
| InfoMax principle | GraphCL | 69.68±0.67 | 73.87±0.66 | 62.40±0.57 | 60.53±0.88 | 75.99±2.65 | 69.80±2.66 | 78.47±1.22 | 75.38±1.44 |
| | AD-GCL | 70.01±1.07 | 76.54±0.82 | 63.07±0.72 | 63.28±0.79 | 79.78±3.52 | 72.30±1.61 | 78.28±0.97 | 78.51±0.80 |
| | JOAOv2 | 71.39±0.92 | 74.27±0.62 | 63.16±0.45 | 60.49±0.74 | 80.97±1.64 | 73.67±1.00 | 77.51±1.17 | 75.49±1.27 |
| | AutoGCL | 73.36±0.77 | 75.69±0.29 | 63.47±0.38 | 62.51±0.63 | 80.99±3.38 | 75.83±1.30 | 78.35±0.64 | 83.26±1.13 |
| Lovász principle (use $\mathcal{L}_{\text{Lo}}$) | GraphCL | 71.37±1.74 | 75.66±1.82 | 63.35±1.47 | 62.11±1.35 | 77.02±1.67 | 72.25±1.42 | 79.23±1.43 | 78.51±1.58 |
| | AD-GCL | 72.24±1.89 | 77.52±1.74 | 63.56±1.36 | 63.87±1.53 | 80.35±2.36 | 74.42±1.57 | 78.95±2.21 | 80.17±1.04 |
| | JOAOv2 | 72.16±1.35 | 75.86±1.21 | 63.92±1.52 | 62.56±1.12 | 81.26±2.37 | 75.94±1.38 | 79.01±2.68 | 79.82±1.39 |
| | AutoGCL | 73.79±1.41 | 76.13±1.48 | 64.21±1.58 | 63.24±1.51 | 81.32±2.12 | 76.04±1.87 | 78.64±1.95 | 82.57±1.95 |
| Lovász principle (use $\mathcal{L}_{\text{ELo}}$) | GraphCL | 73.05±1.21 | 76.45±1.35 | 64.58±1.73 | 63.72±1.52 | 80.21±2.31 | 74.43±1.95 | 80.37±1.52 | 80.63±1.63 |
| | AD-GCL | 72.48±1.59 | 77.96±1.71 | 64.27±1.68 | 63.91±1.74 | 81.76±2.01 | 75.88±1.48 | 81.08±2.35 | 82.21±1.49 |
| | JOAOv2 | 74.13±1.26 | 76.21±1.35 | 64.81±1.92 | 63.38±1.89 | 82.75±2.69 | 76.51±1.53 | 81.13±1.96 | 81.34±1.35 |
| | AutoGCL | 74.67±1.81 | 76.97±1.76 | 65.36±1.45 | 64.13±1.48 | 82.13±2.41 | 76.93±1.62 | 79.56±1.41 | 83.57±1.30 |

## 6.4 Overall Performance and Significance Analysis

For convenience, we show the average performance of all methods over all datasets in Figure 2. We see our Lovász principles outperformed other methods in the three tasks.

To measure the significance of the improvement over the baselines, we implement paired t-tests on the mean scores obtained from the datasets. A p-value below 0.05 indicates a significant difference. The results presented in Table 4 demonstrate the statistical significance of the improvements achieved by our methods across all the datasets.

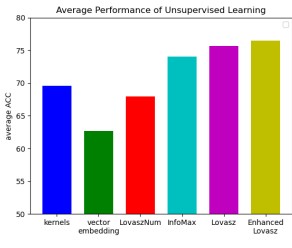

(a) Unsupervised learning

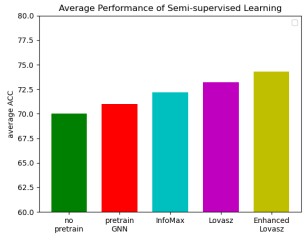

(b) Semi-supervised learning

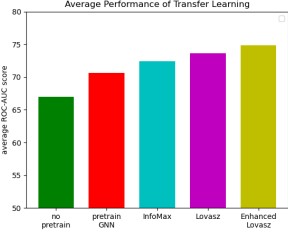

(c) Transfer learning

Figure 2: The average performance of different types of methods

Table 4: Significance analysis ($p$-values) of improvement via the paired t-test. A $p$-value less than 0.05 indicates a significant improvement.

| tasks | principles comparison | InfoGraph | GraphCL | AD-GCL | JOAOv2 | AutoGCL |
|---|---|---|---|---|---|---|
| | InfoMax vs Lovász ($\mathcal{L}_{\text{Lo}}$) | 0.00067 | 0.00286 | 0.02238 | 0.07347 | 0.00059 |
| unsupervised | InfoMax vs Lovász ($\mathcal{L}_{\text{ELo}}$) | 0.00005 | 0.01626 | 0.01541 | 0.01319 | 0.00035 |
| | Lovász ($\mathcal{L}_{\text{Lo}}$) vs Lovász ($\mathcal{L}_{\text{ELo}}$) | 0.00429 | 0.10925 | 0.01522 | 0.00079 | 0.00466 |
| | InfoMax vs Lovász ($\mathcal{L}_{\text{Lo}}$) | - | 0.00028 | 0.01115 | 0.04290 | 0.02147 |
| semi-supervised | InfoMax vs Lovász ($\mathcal{L}_{\text{ELo}}$) | - | 0.00051 | 0.00051 | 0.00116 | 0.01129 |
| | Lovász ($\mathcal{L}_{\text{Lo}}$) vs Lovász ($\mathcal{L}_{\text{ELo}}$) | - | 0.00169 | 0.00076 | 0.00133 | 0.00545 |

## 6.5 Measuring the Quality of Solver Approximation

Given a GNN model $\mathcal{F}_W$ trained via the Lovász principle, the predicted Lovász number of a graph $G$ is denoted as $\hat{\vartheta}(G)$, while the ground-truth Lovász number $\vartheta(G)$ can be computed by SDP [Wolkowicz *et al.*, 2012]. Then we define the relative prediction error for the Lovász number as

$$e_\vartheta = |\hat{\vartheta}(G) - \vartheta(G)|/\vartheta(G). \tag{18}$$

Besides the regularized optimization of the Lovász principle in (9), we also propose a constrained optimization method in Appendix E. We select 50 graphs from each of the four datasets and report $e_\vartheta$ given by both the regularized ($\mu = 10$) optimization and the constrained optimization for Lovász principle in Table 5. We can see that in almost all cases, the relative prediction errors are less than 10%. This indicates that the $\mathcal{F}_W$ trained by the Lovász principle is a good and reliable approximator for the solver $\mathcal{A}$ of the Lovász number. This is similar to the idea of learning to optimize.

Table 5: Relative prediction errors $e_\vartheta$ given by regularized optimization and constrained optimization

| $e_\vartheta$ (%) | MUTAG | PROTEINS | DD | NCI1 |
|---|---|---|---|---|
| regularized optimization | 9.7± 3.4 | 8.2±2.1 | 6.3±1.1 | 10.2± 3.6 |
| constrained optimization | 6.5± 2.4 | 7.3±1.6 | 6.1±1.2 | 8.5± 2.3 |

## 6.6 More Numerical Results

The results of **parameter sensitivity analysis**, **ablation study**, and **time cost comparison** are in Appendix C, Appendix D, and Appendix F respectively.

## 7 Conclusions

This paper proposed a novel method called Lovász principle for unsupervised graph-level representation learning. An extension using the subgraph Lovász number was also presented. The numerical results of unsupervised learning, semi-supervised learning, and transfer learning showed that the proposed methods are more effective than graph kernels and InfoMax principle based representation learning methods. Besides unsupervised representation learning, it is possible to apply our methods to other tasks such as graph-level clustering and graph generation. For instance, we can add a clustering module (e.g. [Xie *et al.*, 2016]) to $\mathcal{L}_{\text{Lo}}$ to construct an end-to-end clustering algorithm. We can combine $\mathcal{L}_{\text{Lo}}$ with variational autoencoder [Kingma and Welling, 2013] to train a model to generate new graphs. Nevertheless, the implementation of these methods is out of the scope of this paper.

## Acknowledgments

This work was partially supported by the Youth program 62106211 of the National Natural Science Foundation of China, the General Program JCYJ20210324130208022 of Shenzhen Fundamental Research, the research funding T00120210002 of Shenzhen Research Institute of Big Data, the Guangdong Key Lab of Mathematical Foundations for Artificial Intelligence, and the funding UDF01001770 of The Chinese University of Hong Kong, Shenzhen.

The authors appreciate the reviewers' and AC's comments and time.

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

# A Comaprision with the equivalent forms of the Lovász principles

In this section, we propose an equivalent form of the Lovász principle based on the complement graph of $G$. We first introduce an equivalent definition of Lovász number as follows.

**Definition A.1** (Lovász number Lovász [1979]). Let $\bar{G}$ be the complement graph of $G$ and $\bar{\mathcal{U}}$ be the orthonormal representations of $\bar{G}$, Lovász proposed an equivalent definition of Lovász number as

$$\vartheta(G) := \max_{\boldsymbol{d}, \boldsymbol{U} \in \bar{\mathcal{U}}} \sum_{p \in V} (\boldsymbol{d}^\top \boldsymbol{u}_p)^2. \tag{19}$$

where $\boldsymbol{d} \in \mathbb{R}^d$ ranges over all unit vectors.

The unit vector $\boldsymbol{d}$ in the Eq. (19) is also a suitable representation vector for graph $G$. Thus we can obtain a Lovász loss $\bar{\mathcal{L}}_{\text{Lo}}$ based on Definition A.1 as follows.

$$\bar{\mathcal{L}}_{\text{Lo}} := \sum_{i=1}^{|\mathcal{G}|} \left( - \sum_{p \in V_i} \left( (\boldsymbol{z}_i^\phi)^\top \boldsymbol{h}_p^\theta \right)^2 \right) + \mu \left( \left\| \bar{\boldsymbol{M}}_i \odot \left( \boldsymbol{H}_i^\theta (\boldsymbol{H}_i^\theta)^\top - \boldsymbol{I}_n \right) \right\|_F^2 + \left( (\boldsymbol{z}_i^\phi)^\top \boldsymbol{z}_i^\phi - 1 \right)^2 \right). \tag{20}$$

where $\bar{\boldsymbol{M}}_i = \boldsymbol{A}_i$ is a mask matrix for complement graph $\bar{G}$. Similar to the main paper, an enhanced Lovász loss based on Definition A.1 at iteration $t$ is defined as

$$\bar{\mathcal{L}}_{\text{ELo}}^{(t)} := \bar{\mathcal{L}}_{\text{Lo}}^{(t)} + \lambda \bar{\mathcal{L}}_{\text{SLN}}^{(t)}, \tag{21}$$

where $\bar{\mathcal{L}}_{\text{SLN}}^{(t)}$ is the subgraph Lovász number (SLN) loss of complement graph $\bar{G}$ based on Lovász number definition (19).

We compare the Lovász loss $\bar{\mathcal{L}}_{\text{Lo}}$ and enhanced Lovász loss $\bar{\mathcal{L}}_{\text{ELo}}^{(t)}$ (based on $\boldsymbol{d}$) with the $\mathcal{L}_{\text{Lo}}$ and $\mathcal{L}_{\text{ELo}}^{(t)}$ (based on *handdle* vector $\boldsymbol{c}$). The results are shown in Tables 6, 7 and 8. For convenience, we show the average performance of all methods over all datasets in Figure 3. We see the two equivalent forms of Lovász principles have very similar performance in the three learning tasks and both of them outperform the InfoMax principle.

Table 6: Performance of unsupervised learning. The **bold**, blue and green numbers denote the best, second best, and third best performances respectively, which also applies to Tables 7 and 8.

| | methods | MUTAG | PROTEINS | DD | NCI1 | COLLAB | IMDB-B | REDDIT-B | REDDIT-M5K |
|---|---|---|---|---|---|---|---|---|---|
| Lovász principle (use $\mathcal{L}_{\text{Lo}}$ and $\boldsymbol{c}$) | InfoGraph | 89.67±1.54 | 75.26±1.43 | 74.13±1.49 | 78.21±1.35 | 71.46±1.21 | 73.87±1.32 | 84.76±1.86 | 54.57±1.38 |
| | GraphCL | 87.24±1.96 | 75.87±2.17 | 79.14±1.67 | 79.13±1.27 | 72.52±1.37 | 72.44±1.46 | 89.87±2.13 | 56.12±1.73 |
| | AD-GCL | 87.44±2.13 | 74.29±2.80 | 76.25±1.48 | 75.12±2.13 | 73.85±1.05 | 73.02±1.35 | 87.11±1.95 | 54.61±2.35 |
| | JOAOv2 | 87.19±1.92 | 73.15±1.46 | 73.15±2.17 | 74.15±1.67 | 72.62±1.43 | 72.18±1.72 | 84.19±1.67 | 53.74±1.70 |
| | AutoGCL | 89.02±1.47 | 76.23±1.25 | 78.95±1.39 | 82.63±2.12 | 71.31±1.72 | 73.95±1.36 | 89.41±1.81 | 57.28±1.62 |
| Lovász principle (use $\mathcal{L}_{\text{ELo}}$ and $\boldsymbol{c}$) | InfoGraph | **90.13±2.05** | 76.12±1.72 | 75.76±1.64 | 79.36±1.57 | 72.67±1.95 | **74.96±1.49** | 84.53±1.79 | 55.12±1.47 |
| | GraphCL | 87.93±2.42 | **76.82±1.34** | 77.35±1.95 | 80.11±1.47 | 74.16±1.37 | 73.87±1.52 | 90.23±1.87 | 56.83±1.35 |
| | AD-GCL | 88.50±1.82 | 75.22±1.93 | 76.14±1.21 | 78.15±1.81 | **74.57±1.98** | 73.48±1.41 | 88.16±1.37 | 55.64±1.63 |
| | JOAOv2 | 88.76±1.43 | 75.27±1.61 | 74.62±2.58 | 76.23±1.75 | 72.85±1.73 | 72.97±1.37 | 85.31±1.48 | 54.68±1.48 |
| | AutoGCL | 89.87±1.85 | 76.03±1.37 | 79.31±1.27 | 82.95±1.26 | 72.23±1.52 | 74.52±1.44 | **90.65±1.46** | 57.93±1.72 |
| Lovász principle (use $\bar{\mathcal{L}}_{\text{Lo}}$ and $\boldsymbol{d}$) | InfoGraph | 88.41±2.47 | 74.37±2.53 | 76.26±1.57 | 77.16±2.43 | 72.51±1.93 | 74.12±1.65 | 86.30±1.92 | 55.18±2.28 |
| | GraphCL | 86.59±1.82 | 76.42±1.97 | 78.35±1.42 | 78.24±1.68 | 71.46±2.02 | 73.81±1.92 | 88.16±1.35 | 54.75±2.41 |
| | AD-GCL | 87.36±1.67 | 75.10±2.33 | 76.77±2.10 | 76.67±1.51 | 74.24±2.17 | 73.14±1.51 | 88.21±2.41 | 55.42±1.31 |
| | JOAOv2 | 88.58±2.30 | 74.26±1.52 | 75.01±1.65 | 77.28±1.92 | 73.21±1.81 | 73.21±1.95 | 85.47±2.02 | 55.09±2.46 |
| | AutoGCL | 88.25±2.11 | 75.78±2.31 | 77.16±1.91 | 80.16±1.07 | 73.11±2.21 | 74.64±2.07 | 87.57±2.14 | 56.83±2.18 |
| Lovász principle (use $\bar{\mathcal{L}}_{\text{ELo}}$ and $\boldsymbol{d}$) | InfoGraph | 89.92±1.26 | 75.37±1.78 | 76.12±1.81 | 80.47±1.99 | 73.18±1.64 | 73.11±1.38 | 85.16±2.31 | 54.87±2.35 |
| | GraphCL | 88.93±2.51 | 75.13±1.26 | 77.52±2.18 | 81.35±1.62 | 73.26±2.01 | 74.21±1.45 | 89.47±2.23 | 55.21±1.92 |
| | AD-GCL | 89.38±1.79 | 76.72±2.09 | 75.33±1.95 | 80.02±2.51 | 74.18±1.44 | 74.09±1.50 | 87.36±1.52 | 56.39±2.32 |
| | JOAOv2 | 87.30±1.25 | 74.62±1.30 | 76.12±1.12 | 79.14±2.50 | 73.11±2.30 | 73.16±1.42 | 87.69±2.20 | 55.67±1.83 |
| | AutoGCL | 89.31±2.01 | 75.23±1.46 | **79.87±1.34** | 82.71±1.34 | 74.20±1.28 | 73.90±1.67 | 90.47±2.52 | **58.17±1.68** |

# B Experimental setting of transfer learning

Following [Hu *et al.*, 2019; You *et al.*, 2021; Yin *et al.*, 2022], we compare the performance of transfer learning of the Lovász principle with the InfoMax principle. We use the Pretrain-GNN method [Hu *et al.*, 2019] as a baseline and employ the Infomax, EdgePred, AttrMasking, and ContextPred pre-training strategies. The experimental settings followed those of AutoGCL [Yin *et al.*, 2022]. Specifically, we performed supervised pre-training for 100 epochs on the ChEMBL dataset [Mayr *et al.*, 2018; Gaulton *et al.*, 2012], and then fine-tuned the model for 30 epochs on 8 chemistry evaluation subsets, using a classifier with a hidden size of 300. Our training employed a batch size of 256 and a learning rate of 0.001. We substitute the InfoMax loss in the four contrastive learning methods (GraphCL, AD-GCL, JOAOv2, and AutoGCL) with the Lovász loss $\mathcal{L}_{\text{Lo}}$ or $\mathcal{L}_{\text{ELo}}$. We repeat each experiment 10 times and report the average ROC-AUC scores in Table 3 of the main paper.

Table 7: Performance of semi-supervised learning.

| | methods | NCI1 | PROTEINS | DD | COLLAB | REDDIT-B | REDDIT-M5K | GITHUB |
|---|---|---|---|---|---|---|---|---|
| Lovász (use $\mathcal{L}_{Lo}$ and $c$) | GraphCL | 75.46±1.53 | 75.12±1.87 | 77.46±1.52 | 76.12±1.15 | 89.87±1.68 | 53.69±1.68 | 66.72±1.53 |
| | AD-GCL | 76.62±1.83 | 74.21±1.71 | 78.27±1.39 | 76.27±1.74 | 90.36±1.56 | 54.06±1.32 | 65.32±1.04 |
| | JOAOv2 | 76.13±1.76 | 73.73±1.86 | 76.27±1.48 | 77.35±1.27 | 89.31±1.85 | 53.17±1.76 | 66.35±1.96 |
| | AutoGCL | 75.77±1.48 | 76.36±1.57 | 78.16±1.61 | 77.63±1.78 | 84.64±2.53 | 51.31±1.81 | 64.87±1.62 |
| Lovász (use $\mathcal{L}_{ELo}$ and $c$) | GraphCL | 75.81±1.68 | 75.88±1.67 | 78.43±1.48 | 77.57±1.58 | 90.67±1.27 | 54.81±1.73 | 67.04±1.45 |
| | AD-GCL | 77.28±1.04 | 75.43±1.58 | 78.67±1.64 | 76.98±1.87 | 91.54±1.39 | 55.46±1.59 | 66.87±1.25 |
| | JOAOv2 | 76.25±1.59 | 74.67±1.37 | 77.96±1.86 | 78.84±1.75 | 90.25±1.22 | 54.32±1.89 | 67.52±1.73 |
| | AutoGCL | 76.53±1.92 | 76.89±1.55 | 78.82±1.90 | 78.46±1.39 | 87.31±1.57 | 53.17±1.50 | 66.47±1.26 |
| Lovász (use $\bar{\mathcal{L}}_{Lo}$ and $d$) | GraphCL | 76.21±1.80 | 76.67±2.14 | 78.20±1.64 | 78.26±2.47 | 90.24±2.13 | 54.71±1.97 | 65.91±2.43 |
| | AD-GCL | 75.71±1.46 | 75.39±1.57 | 77.86±1.77 | 77.53±1.89 | 91.14±2.63 | 54.17±2.75 | 66.14±1.71 |
| | JOAOv2 | 76.40±2.32 | 74.61±1.28 | 77.26±2.01 | 76.45±1.34 | 88.27±1.89 | 54.34±2.53 | 65.41±2.76 |
| | AutoGCL | 76.84±1.53 | 76.49±2.03 | 78.35±1.75 | 78.02±2.41 | 86.17±1.61 | 52.26±1.76 | 64.26±2.41 |
| Lovász (use $\bar{\mathcal{L}}_{ELo}$ and $d$) | GraphCL | 76.28±1.59 | 76.21±2.30 | 79.34±2.51 | 78.10±1.76 | 91.21±2.04 | 55.62±2.68 | 66.59±1.62 |
| | AD-GCL | 77.65±2.17 | 75.52±1.97 | 78.71±1.93 | 77.29±2.33 | 90.76±2.18 | 55.14±1.73 | 67.31±2.41 |
| | JOAOv2 | 76.19±2.30 | 75.41±2.24 | 78.03±2.58 | 79.31±1.49 | 91.34±2.11 | 53.70±2.41 | 66.35±2.28 |
| | AutoGCL | 75.42±1.87 | 77.25±2.67 | 77.54±1.26 | 78.27±2.43 | 88.69±2.63 | 54.28±1.64 | 66.19±1.67 |

Table 8: Performance of transfer learning.

| | methods | BBBP | Tox21 | ToxCast | SIDER | ClinTox | MUV | HIV | BACE |
|---|---|---|---|---|---|---|---|---|---|
| Lovász (use $\mathcal{L}_{Lo}$ and $c$) | GraphCL | 71.37±1.74 | 75.66±1.82 | 63.35±1.47 | 62.11±1.35 | 77.02±1.67 | 72.25±1.42 | 79.23±1.43 | 78.51±1.58 |
| | AD-GCL | 72.24±1.89 | 77.52±1.74 | 63.56±1.36 | 63.87±1.53 | 80.35±2.36 | 74.42±1.57 | 78.95±2.21 | 80.17±1.04 |
| | JOAOv2 | 72.16±1.35 | 75.86±1.21 | 63.92±1.52 | 62.56±1.12 | 81.26±2.37 | 75.94±1.38 | 79.01±2.68 | 79.82±1.39 |
| | AutoGCL | 73.79±1.41 | 76.13±1.48 | 64.21±1.58 | 63.24±1.51 | 81.32±2.12 | 76.04±1.87 | 78.64±1.95 | 82.57±1.95 |
| Lovász (use $\mathcal{L}_{ELo}$ and $c$) | GraphCL | 73.05±1.21 | 76.45±1.35 | 64.58±1.73 | 63.72±1.52 | 80.21±2.31 | 74.43±1.95 | 80.37±1.52 | 80.63±1.63 |
| | AD-GCL | 72.48±1.59 | 77.96±1.71 | 64.27±1.68 | 63.91±1.74 | 81.76±2.01 | 75.88±1.48 | 81.08±2.35 | 82.21±1.49 |
| | JOAOv2 | 74.13±1.26 | 76.21±1.35 | 64.81±1.92 | 63.38±1.89 | 82.75±2.69 | 76.51±1.53 | 81.13±1.96 | 81.34±1.35 |
| | AutoGCL | 74.67±1.81 | 76.97±1.76 | 65.36±1.45 | 64.13±1.48 | 82.13±2.41 | 76.93±1.62 | 79.56±1.41 | 83.57±1.30 |
| Lovász (use $\bar{\mathcal{L}}_{Lo}$ and $d$) | GraphCL | 72.53±2.16 | 75.27±1.35 | 64.20±1.82 | 63.16±2.25 | 79.17±1.63 | 74.53±2.28 | 78.61±1.55 | 80.43±1.77 |
| | AD-GCL | 71.45±1.77 | 76.48±2.03 | 63.78±2.21 | 62.85±2.31 | 79.33±2.49 | 73.19±1.22 | 79.42±1.43 | 78.26±1.47 |
| | JOAOv2 | 71.78±2.27 | 75.19±1.87 | 64.26±1.65 | 63.27±1.80 | 81.46±1.83 | 74.39±2.48 | 79.59±2.17 | 80.14±2.51 |
| | AutoGCL | 72.52±2.30 | 76.46±2.33 | 63.81±2.53 | 62.84±1.34 | 80.91±2.27 | 75.57±2.16 | 79.35±1.42 | 81.66±1.60 |
| Lovász (use $\bar{\mathcal{L}}_{ELo}$ and $d$) | GraphCL | 74.54±2.23 | 76.32±2.54 | 65.62±1.74 | 64.34±1.86 | 81.75±1.61 | 75.20±2.31 | 81.64±2.37 | 81.27±1.58 |
| | AD-GCL | 71.74±2.81 | 77.85±1.95 | 64.81±2.52 | 64.25±2.57 | 80.54±1.32 | 75.34±1.76 | 80.56±1.64 | 81.65±2.48 |
| | JOAOv2 | 73.71±1.68 | 76.43±1.64 | 64.92±2.17 | 63.56±1.67 | 81.96±1.58 | 76.78±1.84 | 81.53±2.42 | 80.76±2.31 |
| | AutoGCL | 74.69±2.10 | 76.21±1.81 | 65.13±1.83 | 63.77±2.52 | 82.47±2.52 | 75.42±2.25 | 80.74±1.56 | 82.89±2.68 |

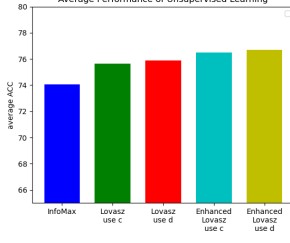

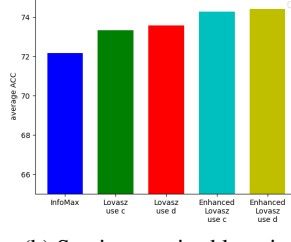

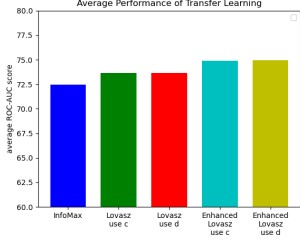

(a) Unsupervised learning     (b) Semi-supervised learning     (c) Transfer learning

Figure 3: The average performance of different types of methods

## C   Parameter sensitivity analysis

There are five hyperparameters need to be tuned in Lovász principle method: the dimension of representations $d$, the hyperparameter of orthonormal representation regularization $\mu$, the hyperparameter of orthogonal regularization in subgraph Lovász number (SLN) loss $\gamma$, hyperparameter of subgraph Lovász number (SLN) loss in enhanced Lovász loss $\eta$, the hyperparameter of graph-level representations $\ell_2$-norm regularization in semi-supervised Lovász loss $\lambda$. In this section, we analyze the parameter sensitivity on the InfoGraph Sun *et al.* [2019] with different hyperparameters. We repeat each experiment for ten times and plot the average accuracy on different datasets.

### C.1   $d$ **as the dimension of representations**

In Lovász principle, $d$ is the dimension of node-level representations $\boldsymbol{H} = [\boldsymbol{h}_1, ..., \boldsymbol{h}_n]^\top \in \mathbb{R}^{n \times d}$ and graph-level representation of $G$, $\boldsymbol{h}_v \in \mathbb{R}^d$. In the definition of Lovász number, the node-level representations are orthonormal representations, i.e., $\boldsymbol{H}^\top \in \mathcal{U}$. Let $\alpha(G)$ be the independent number of graph $G$, which is the size of the maximum independent set. If $d \leq \alpha(G)$, the node-level representations $\boldsymbol{H}$ are impossible to be orthonormal representations such that the Lovász number cannot be obtained. In Figure 4, we fix other hyperparameters and tune $d$ from $\{10, 20, ..., 90, 100\}$.

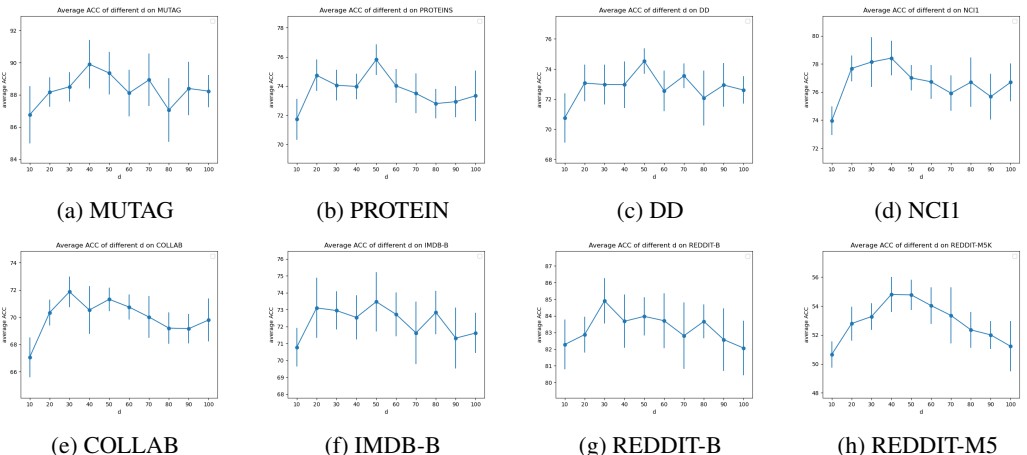

Figure 4: The average ACC of different $d$ on different dataset

The results show that a small $d$ adversely affects the performance because $d$ may be less than $\alpha(G)$ on some graphs. When $d$ is too large, the average accuracy decreases slightly because the representations with large $d$ may capture some noisy information of a graph.

## C.2 $\mu$ for orthonormal representation regularization

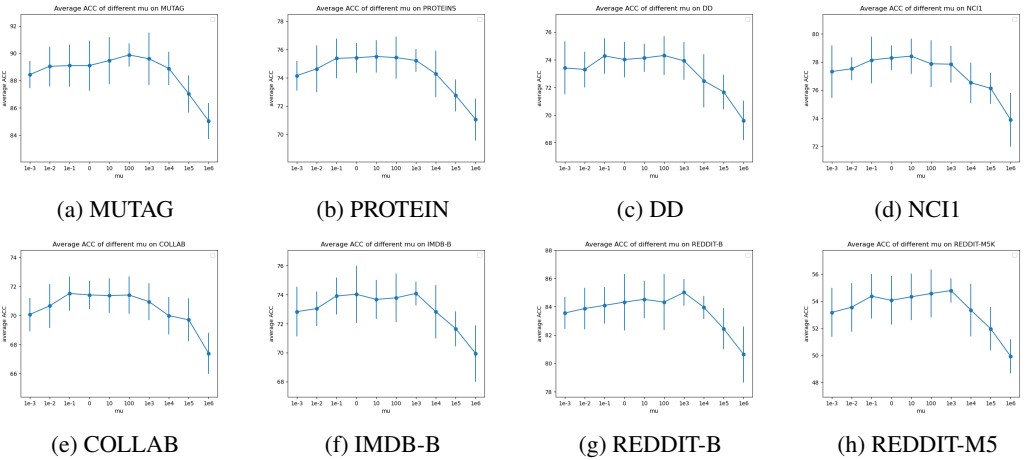

Figure 5: The average ACC of different $\mu$ on different data

In Lovász principle, $\mu$ is the hyperparameter for orthonormal representation regularization in Lovász loss $\mathcal{L}_{\mathrm{Lo}}$ (9) as follows

$$\mathcal{L}_{\mathrm{Lo}} := \sum_{i=1}^{|\mathcal{G}|} \max_{p \in V_i} \frac{1}{((\boldsymbol{z}_i^\phi)^\top \boldsymbol{h}_p^\theta)^2} + \mu \left( \left\| \boldsymbol{M}_i \odot \left( \boldsymbol{H}_i^\theta (\boldsymbol{H}_i^\theta)^\top - \boldsymbol{I}_n \right) \right\|_F^2 + \left( (\boldsymbol{z}_i^\phi)^\top \boldsymbol{z}_i^\phi - 1 \right)^2 \right). \quad (22)$$

In Figure 5, we fix other hyperparameters and tune $\mu$ from $\{10^{-3}, 10^{-2}, ..., 10^5, 10^6\}$. The results show that $\mu$ is not sensitive when $0.1 \leq \mu \leq 1e3$. If $\mu$ is too small, the average accuracy decreases slightly because the node-level representations $\boldsymbol{H}$ may not be orthonormal representations. A very large $\mu$ adversely affects the performance because the orthonormal representation regularization dominates the representation learning such that the Lovász principle fails.

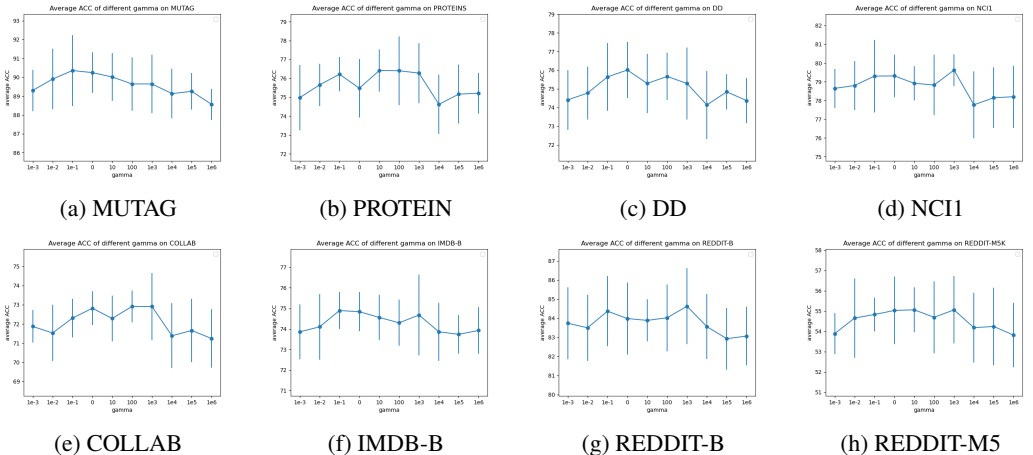



(a) MUTAG      (b) PROTEIN      (c) DD      (d) NCI1

(e) COLLAB      (f) IMDB-B      (g) REDDIT-B      (h) REDDIT-M5



Figure 6: The average ACC of different $\gamma$ on different data

## C.3 $\gamma$ for orthogonal regularization in subgraph Lovász number (SLN) loss

In Lovász principle, $\gamma$ is the hyperparameter for orthogonal regularization in subgraph Lovász number (SLN) loss $\mathcal{L}_{\text{SLN}}^{(t)}$ as follows

$$\mathcal{L}_{\text{SLN}}^{(t)} := \sum_{i=1}^{|\mathcal{G}|} \sum_{j=1}^{|\mathcal{G}|} K_{ij}^{(t-1)} \|\boldsymbol{z}_i^\phi - \boldsymbol{z}_j^\phi\|_2^2 + \gamma(\|\boldsymbol{Z}_\phi^\top \boldsymbol{Z}_\phi - \boldsymbol{I}_d\|_F^2 + \|\boldsymbol{Z}_\phi^\top \boldsymbol{1}_{N \times 1}\|_2^2), \tag{23}$$

In Figure 6, we fix other hyperparameters and tune $\gamma$ from $\{10^{-3}, 10^{-2}, ..., 10^5, 10^6\}$. The results show that $\gamma$ is not sensitive when $0.1 \le \gamma \le 1e3$. If $\gamma$ is too small, the average accuracy decreases slightly because the orthogonal constraints of spectral embedding may not hold. A very large $\gamma$ adversely affects the performance because the orthogonal regularization of spectral embedding dominates the representation learning such that the Lovász principle fails.

## C.4 $\eta$ for subgraph Lovász number (SLN) loss in enhanced Lovász loss

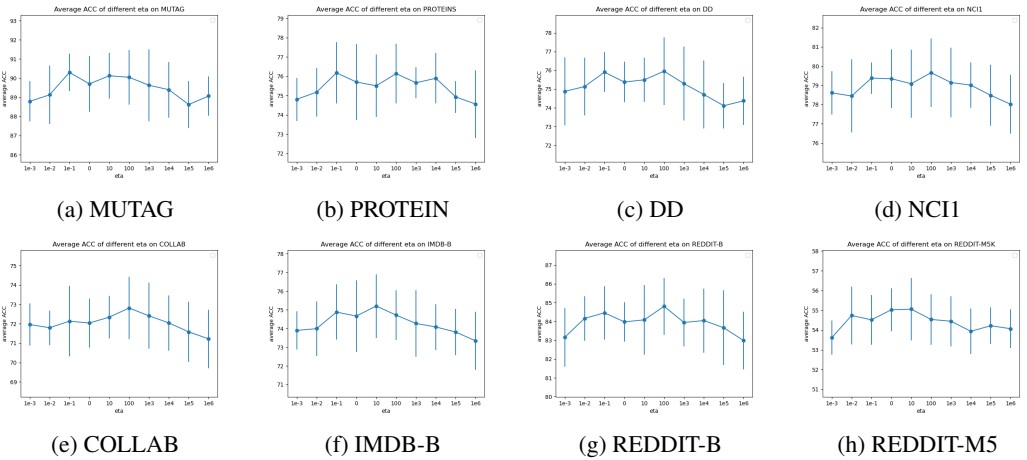



(a) MUTAG      (b) PROTEIN      (c) DD      (d) NCI1

(e) COLLAB      (f) IMDB-B      (g) REDDIT-B      (h) REDDIT-M5



Figure 7: The average ACC of different $\eta$ on different data

In Lovász principle, $\eta$ is the hyperparameter for subgraph Lovász number (SLN) loss $\mathcal{L}_{\text{SLN}}^{(t)}$ in enhanced Lovász loss $\mathcal{L}_{\text{ELo}}^{(t)}$ as follows

$$\mathcal{L}_{\text{ELo}}^{(t)} := \mathcal{L}_{\text{Lo}}^{(t)} + \eta \mathcal{L}_{\text{SLN}}^{(t)}.$$

In Figure 7, we fix other hyperparameters and tune $\eta$ from $\{10^{-3}, 10^{-2}, ..., 10^5, 10^6\}$. The results show that $\eta$ is not sensitive when $10^{-2} \leq \eta \leq 1e4$. If $\eta$ is a very small number, the enhanced Lovász loss $\mathcal{L}_{\text{ELo}}^{(t)}$ degenerates into the Lovász loss $\mathcal{L}_{\text{Lo}}$, which also performs well in representation learning. A very large $\eta$ adversely affects the performance because the subgraph Lovász number (SLN) loss $\mathcal{L}_{\text{SLN}}^{(t)}$ dominates the representation learning such that the Lovász principle may fail.

## C.5 $\quad \lambda$ for the $\ell_2$-norm regularization in semi-supervised Lovász loss

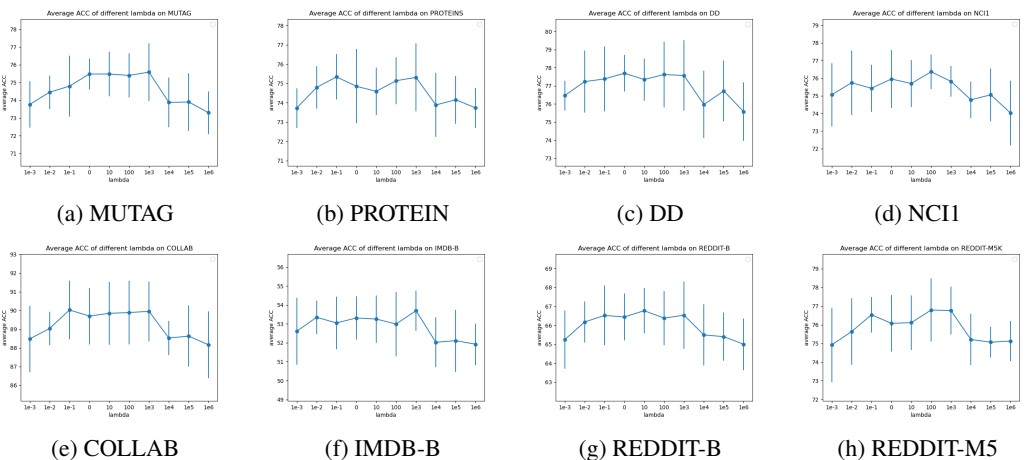

|   |   |   |   |
|---|---|---|---|
| (a) MUTAG | (b) PROTEIN | (c) DD | (d) NCI1 |
| (e) COLLAB | (f) IMDB-B | (g) REDDIT-B | (h) REDDIT-M5 |

Figure 8: The average ACC of different $\lambda$ on different data

In semi-supervised learning, $\lambda$ the hyperparameter for the $\ell_2$-norm regularization in semi-supervised Lovász loss as follows

$$\mathcal{L}_{\text{Lo-semi}} := \sum_{l=1}^{|\mathcal{G}^L|} \mathcal{L}_{\text{supervised}}(\hat{\boldsymbol{y}}_l^\psi, \boldsymbol{y}_l) + \mathcal{L}_{\text{unsupervised}}(\boldsymbol{H}_i^\theta, \boldsymbol{z}_i^\phi) + \lambda \sum_{i=1}^{|\mathcal{G}|} \|\boldsymbol{z}_i^\phi - \boldsymbol{z}_i^\psi\|_2^2, \qquad (24)$$

In Figure 8, we fix other hyperparameters and tune $\lambda$ from $\{10^{-3}, 10^{-2}, ..., 10^5, 10^6\}$. The results show that $\eta$ is not sensitive when $0.1 \leq \gamma \leq 1e3$. If $\lambda$ is a very small number, the supervised encoder and unsupervised encoder may learn different information of a graph $G$ such that the average accuracy slightly decreases. A very large $\eta$ will cause the training to be trapped in early iterations such that the representation learning fails.

# D  Ablation study

In this section, we analyze the importance of the orthonormal representation regularization (Eq. (9)) and the subgraph Lovász number (SLN) loss (Eq. (14)) by ablation study on unsupervised learning.

## D.1  Ablation study of orthonormal representation regularization

In the ablation study, we remove the orthonormal representation regularization in Eq. (9) by setting $\mu = 0$. The results in Table 9 show that the orthonormal representation regularization can improve the performance of graph representation learning for two reasons:

- The orthonormal representation constraint is part of the definition of Lovász number such that the Lovász principle may fail without the orthonormal representation regularization.

- The orthonormal representation regularization guides the method the learn the structure property of a graph.

Table 9: Performance (ACC) of unsupervised learning for Ablation study. The ablation indicates $\mu = 0$ in Eq. (9). The **bold** numbers denote the better performances of the same method.

| | methods | ablation | MUTAG | PROTEINS | DD | NCI1 | COLLAB | IMDB-B | REDDIT-B | REDDIT-M5K |
|---|---|---|---|---|---|---|---|---|---|---|
| Lovász principle (use $\mathcal{L}_{Lo}$) | InfoGraph | ✓ | 88.25±1.63 | 74.38±1.52 | 73.05±1.77 | 76.89±1.42 | **72.23±1.35** | 72.09±1.41 | 83.56±1.22 | 53.21±1.90 |
| | | × | **89.67±1.54** | **75.26±1.43** | **74.13±1.49** | **78.21±1.35** | 71.46±1.21 | **73.87±1.32** | **84.76±1.86** | **54.57±1.38** |
| | GraphCL | ✓ | 86.16±1.31 | 73.48±1.56 | 78.22±1.09 | 77.84±1.73 | 71.65±1.20 | 71.82±1.78 | 87.79±1.86 | 55.03±2.17 |
| | | × | **87.24±1.96** | **75.87±2.17** | **79.14±1.67** | **79.13±1.27** | **72.52±1.37** | **72.44±1.46** | **89.87±2.13** | **56.12±1.73** |
| | AD-GCL | ✓ | 86.28±1.84 | 73.52±1.78 | 75.89±1.67 | 74.53±1.36 | 72.41±1.58 | 72.84±1.52 | **88.46±1.35** | 53.27±1.51 |
| | | × | **87.44±2.13** | **74.29±2.80** | **76.25±1.48** | **75.12±2.13** | **73.85±1.05** | **73.02±1.35** | 87.11±1.95 | **54.61±2.35** |
| | JOAOv2 | ✓ | **88.34±1.36** | 72.49±1.19 | 72.53±2.07 | 73.64±2.52 | 72.27±1.59 | 71.63±1.81 | 83.48±1.31 | 53.02±1.21 |
| | | × | 87.19±1.92 | **73.15±1.46** | **73.15±2.17** | **74.15±1.67** | **72.62±1.43** | **72.18±1.72** | **84.19±1.67** | **53.74±1.70** |
| | AutoGCL | ✓ | 88.76±2.15 | 75.59±2.03 | 78.42±1.81 | 81.79±1.65 | 70.42±1.30 | 73.64±1.52 | 88.31±1.22 | 56.53±1.27 |
| | | × | **89.02±1.47** | **76.23±1.25** | **78.95±1.39** | **82.63±2.12** | **71.31±1.72** | **73.95±1.36** | **89.41±1.81** | **57.28±1.62** |
| Lovász principle (use $\mathcal{L}_{ELo}$) | InfoGraph | ✓ | 89.51±1.63 | 75.18±1.67 | **76.29±1.52** | 78.24±1.61 | 71.22±1.43 | 74.35±1.07 | 83.87±2.39 | **56.23±1.15** |
| | | × | **90.13±2.05** | **76.12±1.72** | 75.76±1.64 | **79.36±1.57** | **72.67±1.95** | **74.96±1.49** | **84.53±1.79** | 55.12±1.47 |
| | GraphCL | ✓ | 86.78±1.57 | 76.24±1.60 | 76.71±1.63 | 79.42±1.03 | 73.37±1.26 | 73.42±1.07 | 89.70±1.52 | 55.64±1.25 |
| | | × | **87.93±2.42** | **76.82±1.34** | **77.35±1.95** | **80.11±1.47** | **74.16±1.37** | **73.87±1.52** | **90.23±1.87** | **56.83±1.35** |
| | AD-GCL | ✓ | 88.36±1.51 | 74.63±1.74 | **76.39±1.37** | 77.68±1.23 | 74.03±1.52 | 72.73±1.22 | 87.31±1.21 | **55.87±1.42** |
| | | × | **88.50±1.82** | **75.22±1.93** | 76.14±1.21 | **78.15±1.81** | **74.57±1.98** | **73.48±1.41** | **88.16±1.37** | 55.64±1.63 |
| | JOAOv2 | ✓ | 88.27±1.39 | 74.52±1.40 | 74.31±1.15 | 75.78±1.43 | 72.10±1.62 | 72.39±2.15 | 84.45±1.36 | **54.21±1.52** |
| | | × | **88.76±1.43** | **75.27±1.61** | **74.62±2.58** | **76.23±1.75** | **72.85±1.73** | **72.97±1.37** | **85.31±1.48** | 54.68±1.48 |
| | AutoGCL | ✓ | 89.24±2.43 | 75.29±1.58 | 78.42±1.68 | 82.31±1.79 | 71.69±1.98 | **74.46±1.65** | 90.10±1.73 | 57.27±1.54 |
| | | × | **89.87±1.85** | **76.03±1.37** | **79.31±1.27** | **82.95±1.26** | **72.23±1.52** | 74.52±1.44 | **90.65±1.46** | **57.93±1.72** |

## D.2 Ablation study of subgraph Lovász number (SLN) loss

If we remove the subgraph Lovász number (SLN) loss in Eq. (14) by setting $\eta = 0$, the enhanced Lovász loss (14) degenerates to simple Lovász loss (9). The results in Table 10 show that the subgraph Lovász number (SLN) loss can improve the performance of graph representation learning because similar graphs are guaranteed to be close to each other in the representation space.

## D.3 Ablation study of orthonormal representation regularization

Table 10: Performance (ACC) of unsupervised learning for Ablation study. The ablation indicates $\eta = 0$ in Eq. (14). The **bold** numbers denote the better performances of the same method.

| methods | ablation | MUTAG | PROTEINS | DD | NCI1 | COLLAB | IMDB-B | REDDIT-B | REDDIT-M5K |
|---|---|---|---|---|---|---|---|---|---|
| InfoGraph | ✓ | 89.67±1.54 | 75.26±1.43 | 74.13±1.49 | 78.21±1.35 | 71.46±1.21 | 73.87±1.32 | **84.76±1.86** | 54.57±1.38 |
| | × | **90.13±2.05** | **76.12±1.72** | **75.76±1.64** | **79.36±1.57** | **72.67±1.95** | **74.96±1.49** | 84.53±1.79 | **55.12±1.47** |
| GraphCL | ✓ | 87.24±1.96 | 75.87±2.17 | **79.14±1.67** | 79.13±1.27 | 72.52±1.37 | 72.44±1.46 | 89.87±2.13 | 56.12±1.73 |
| | × | **87.93±2.42** | **76.82±1.34** | 77.35±1.95 | **80.11±1.47** | **74.16±1.37** | **73.87±1.52** | **90.23±1.87** | **56.83±1.35** |
| AD-GCL | ✓ | 87.44±2.13 | 74.29±2.80 | **76.25±1.48** | 75.12±2.13 | 73.85±1.05 | 73.02±1.35 | 87.11±1.95 | 54.61±2.35 |
| | × | **88.50±1.82** | **75.22±1.93** | 76.14±1.21 | **78.15±1.81** | **74.57±1.98** | **73.48±1.41** | **88.16±1.37** | **55.64±1.63** |
| JOAOv2 | ✓ | 87.19±1.92 | 73.15±1.46 | 73.15±2.17 | 74.15±1.67 | 72.62±1.43 | 72.18±1.72 | 84.19±1.67 | 53.74±1.70 |
| | × | **88.76±1.43** | **75.27±1.61** | **74.62±2.58** | **76.23±1.75** | **72.85±1.73** | **72.97±1.37** | **85.31±1.48** | **54.68±1.48** |
| AutoGCL | ✓ | 89.02±1.47 | **76.23±1.25** | 78.95±1.39 | 82.63±2.12 | 71.31±1.72 | 73.95±1.36 | 89.41±1.81 | 57.28±1.62 |
| | × | **89.87±1.85** | 76.03±1.37 | **79.31±1.27** | **82.95±1.26** | **72.23±1.52** | **74.52±1.44** | **90.65±1.46** | **57.93±1.72** |

# E Strict Lovász Principle

We use regularization in Lovász principle (9) instead of constraint because its optimization is much easier and its performance is very close to the constrained optimization. In Algorithm 1, we propose a constrained optimization for the "strict Lovász principle" via projection.

---

**Algorithm 1:** Constrained optimization for "strict Lovász principle"

---
1: **Initialization:** $\mu = 1$
2: **repeat**
3:     $\hat{H}^t = F(A, X; \theta^t)$ and $\hat{z}^t = f(A, X; \phi^t)$
4:     $H^t = \text{Proj}_U(\hat{H}^t)$ and $z^t = \frac{\hat{z}^t}{\|\hat{z}^t\|}$
5:     obtain $\theta^{t+1}, \phi^{t+1}$ by SGD updating
6: **until** Convergence

---

In Algorithm 1, the $\text{Proj}_U$ project $\hat{H}^t$ to the orthonormal representation space, which is similar to the Gram–Schmidt process. We define $\text{proj}_w(h) := \frac{\langle h, w \rangle}{\langle w, w \rangle} w$. Let $W_k$ be the set that $W_k := \{w_1, w_2, ..., w_k\}$. For each vertex $i \in V$, we denote $\Omega_i$ as the set of vector $w_j$s where $j$ can be each vertex not adjacent to vertex $i$. Then the $H^t = \text{Proj}_U(\hat{H}^t)$ is defined in Algorithm 2

**Algorithm 2:** The definition of projection function $\text{Proj}_U$

---

1: $w_1 = \hat{h}_1^t, \ e_1 = \frac{w_1}{\|w_1\|}$

2: $w_2 = \hat{h}_2^t - \sum_{w \in W_{2-1} \cap \Omega_2} \text{proj}_w(\hat{h}_2^t), \ e_2 = \frac{w_2}{\|w_2\|}$

3: ...

4: $w_k = \hat{h}_k^t - \sum_{w \in W_{k-1} \cap \Omega_k} \text{proj}_w(\hat{h}_k^t), \ e_k = \frac{w_k}{\|w_k\|}$

5: ...

6: $w_n = \hat{h}_n^t - \sum_{w \in W_{n-1} \cap \Omega_n} \text{proj}_w(\hat{h}_n^t), \ e_n = \frac{w_n}{\|w_n\|}$

7: Output $H^{t+1} = [e_1, e_2, ..., e_n]^\top$

---

The comparisons between the regularized ($\mu = 1$) optimization and the constrained optimization for two methods on four datasets are as follows.

Table 11: Comparison between regularized optimization and constrained optimization

| | method | MUTAG | PROTEINS | DD | NCI1 |
|---|---|---|---|---|---|
| regularized opt. | InfoGraph | 89.67± 1.54 | 75.26 ± 1.43 | 74.13± 1.49 | 78.21± 1.35 |
| regularized opt. | GraphCL | 87.24±1.96 | 75.87± 2.17 | 79.14 ± 1.67 | 79.13± 1.27 |
| constrained opt. | InfoGraph | 86.12± 2.32 | 75.49± 1.52 | 76.42± 1.56 | 77.80± 1.24 |
| constrained opt. | GraphCL | 87.52 ± 2.75 | 76.11± 1.36 | 78.54 ± 2.21 | 77.63± 1.58 |

## F   Time cost comparison

We compare the time cost between the InfoMax principle and Lovász principle using the InfoGraph Sun *et al.* [2019] model on different datasets. We run the programming on a machine with Intel 7 CPU and RTX 3090 GPU. We repeat the experiment five times and report the results in Table 12. The Lovász principle is the fastest method among the three. The Lovász principle is the fastest method

Table 12: Time cost of InfoGraph. h stands for hour and m stands for minute. The brown value indicates the lowest time cost.

| tasks | principle | MUTAG | PROTEINS | DD | NCI1 | COLLAB | IMDB-B | REDDIT-B | REDDIT-M5K |
|---|---|---|---|---|---|---|---|---|---|
| unsupervised learning | InfoMax | 2.3 m | 12.6 m | 1 h 39 m | 36.5 m | 1 h 50 m | 5.8 m | 3 h 14 m | 7 h 31 m |
| | Lovász | 1.8 m | 11.7 m | 1 h 25 m | 33.2 m | 1 h 46 m | 5.1 m | 3 h 9 m | 7 h 26 m |
| | Enhanced Lovász | 1.8 m | 12.2 m | 1 h 30 m | 34.3 m | 1 h 47 m | 5.3 m | 3 h 10 m | 7 h 27 m |
| semi-supervised learning | InfoMax | 2.3 m | 13.1 m | 1 h 47 m | 46.3 m | 2 h 21 m | 9.6 m | 3 h 47 m | 8 h 52 m |
| | Lovász | 2.0 m | 12.7 m | 1 h 39 m | 43.1 m | 2 h 15 m | 8.1 m | 3 h 27 m | 8 h 16 m |
| | Enhanced Lovász | 2.0 m | 12.8 m | 1 h 40 m | 42.6 m | 2 h 17 m | 8.3 m | 3 h 29 m | 8 h 20 m |

among the three, and the reasons are as follows:

- The Lovász principle is faster than the InfoMax principle because the former does not use the $f$-GAN [Nowozin *et al.*, 2016] and the Jensen-Shannon MI estimator $I_\varphi$ to evaluate the mutual information.

- Enhanced Lovász principle is slightly lower than Lovász principle because the computation of $1/(\boldsymbol{z}_i^{(t-1)\top} \boldsymbol{u}_p^{(t-1)})^2$ for every $p \in V_i$ was already done when computing $\mathcal{L}_{\text{Lo}}$.

