# OpenReview forum: "Lovász Principle for Unsupervised Graph Representation Learning"
_NeurIPS.cc/2023/Conference — NeurIPS 2023 poster_

### Official Review · Reviewer_2QoV · 2023-06-26

**Soundness:** 4 excellent
**Presentation:** 4 excellent
**Contribution:** 3 good
**Rating:** 7
**Confidence:** 4

**Summary:**

The paper introduces the "Lovasz Principle", an unsupervised or semi-supervised graph representation learning approach inspired by the Lovasz number. The motivation for using the Lovasz Principle is well established, and the authors extensively discuss related work and similar approaches. The experimental setup is sound, and the Lovasz Principle is tested against many widely used graph representation learners, with better results in all experiments.


**Strengths:**

- The paper is well-written and easy to follow, with compelling arguments and intuitive explanations.

- The steps taken to derive the Lovasz Principle are well laid on.

- The experimental setup is sound and contains many different approaches to graph representation learning. Variants of the Lovasz Principle consistently obtain the best scores on all datasets. I appreciate the inclusion of Figure 2 for a quick visual summary.

- Compared to InfoMax, the Lovasz Principle is simpler because it does not require a discriminator for the Jensen-Shannon MI estimator. The Lovasz Principle also has a faster runtime.


**Weaknesses:**

- The authors discuss the non-uniqueness of $U_i*$ and $c_i*$ and give some intuition regarding why this is the case, but mention that they "hope that similar graphs have similar representations". It would be interesting and make the argument more compelling if the authors computed $c*$ for various "similar" and "different" graphs and showed empirically that they are indeed similar or different.

- Similarly, it would be interesting if the authors empirically showed that $\mathcal{F}_W$ is actually a good approximator for $(U_i*, c_i*)$ by comparing them with $\mathcal{A}(G_i)$, where $\mathcal{A}$ is some solver.


- It would make the paper more readable if the authors moved the footnote from page 4 to the caption of Tab. 1.

- I appreciate that the authors go into great detail regarding hyperparameter sensitivity in the Supplementary material, but they don't mention the effect of the hyperparameters in the main paper. The reader should be aware that hyperparameters can significantly affect the final performance, and one should consider this when using the Lovasz Principle.

- I could not find any code provided by the authors for reproducibility.

**Questions:**

- While Figure 1 and the description for the pentagon example are great, I was slightly confused when I initially read the part regarding the orthogonal vector pairs. This is because I've read $(u_i*, u_j*)$ as being a vector, not a pair. It would be great if the authors somehow clarified the intuition about the orthogonality.

- I assume that $\odot$ in Eq. 8 is the Hadamard product? It would be nice to state this somewhere explicitly.

- The authors mention obtaining node-level representations via the $F(\cdot, \cdot, \theta)$ function but are tackling only graph-level tasks. Have you tried node-level tasks? It would be interesting to see the approach tested on them.

- Have you tried assessing if $\mathcal{F}_W$ is a good approximator for $(U_i*, c_i*)$? Maybe the formulation helps only in a representation-learning sense and doesn't produce a good approximator.

- Will the code be publicly available? I will update my score accordingly if this is the case.

**Limitations:**

- The Lovasz Principle adds some new hyperparameters that could affect the final performance if not correctly chosen. The authors have included an extensive discussion regarding this in the Supplementary to the paper, but I believe this should also be mentioned in the main paper.

---

> ### Author Rebuttal · Authors · 2023-08-08
>
> **Weakness 1: Handle vector comparison between "similar" and "different" graphs**
>
> **Response:** Given two graph sets $\mathcal{S}_ 1$ and $\mathcal{S}_ 2$, we denote $c_ i$ as the handle vector for $G_ i$, $i=1,2$. We define the following cross-set difference:
>
> $\ell_ {c}(\mathcal{S}_ 1, \mathcal{S}_ 2) =  \frac{1}{|\mathcal{S}_ 1| \times |\mathcal{S}_ 2|} \sum_{G_ i \in \mathcal{S}_ 1, G_ j \in \mathcal{S}_ 2} \Vert z_ i - z_ j \Vert_ 2^2.$
>
> Given a graph dataset $\mathcal{G}$, it is natural to assume that the graphs in the same class are "similar" and graphs in the different class are "different". Then we select $\mathcal{S}_1$ and $\mathcal{S}_2$ from the first class of $\mathcal{G}$ where $\mathcal{S}_1 \cap \mathcal{S}_2 = \emptyset$. $\mathcal{S}_3$ is selected from the second class of $\mathcal{G}$. We use InfoGraph as our GNN framework by substituting its InfoMax loss with the Lovasz loss Eqn. (9). For each dataset, we select 30 graphs for each subset. We repeat the experiment10 times and report the results as follows:
>
> \begin{matrix}
> \hline
>  Difference & MUTAG &  PROTEINS &  DD & NCI1  \\\\     \hline
>  \ell_{c}(\mathcal{S}_ 1, \mathcal{S}_ 2) &  0.24\pm 0.13   & 0.27 \pm 0.08  &0.23 \pm 0.10 &0.26 \pm 0.12  \\\\
> \ell_{c}(\mathcal{S}_ 1, \mathcal{S}_ 3) &  0.31\pm 0.16   & 0.33 \pm 0.11  &0.28 \pm 0.15 &0.35 \pm 0.17  \\\\ \hline
> \end{matrix}
>
> The results show that the handle vector of "similar" graphs are close, while that of "different" graphs are different.
>
> **Weakness 2: $\mathcal{F}_W$ vs $\mathcal{A}(G)$**
>
> **Response: please refer to our response to your Question 4.**
>
> **Question 1: intuition about orthogonality**
>
> **Response:**  In [1], a coding source with symbols is represented as a graph $G = (V, E)$, where each vertex represents one symbol. If the symbols of vertex $i$ and vertex $j$ share some information in common, these two symbols are probably confused in coding a message. This confusable relationship is represented as an edge between vertex $i$ and vertex $j$. Laszlo Lovasz proposed to represent the information of a coding symbol as an orthonormal vector. If vertex $i$ and vertex $j$ are not adjacent, that means they share no information in common, and thus their vector representations $v_i$ and $v_j$ should be orthogonal to each other, thus $v_i^\top v_j = 0$. That is the intuition about the orthogonality, which is derived from the information theory view.
>
> [1] László Lovász. On the Shannon capacity of a graph
>
> **Question 2:** $\odot$ is a Hadamard product**.
>
> **Response:** Yes. Will state it explicitly.
>
> **Qeustion 3: Lovasz principle for Node-level tasks**
>
> **Response:** Yes. Our Lovasz principle can also be applied to node-level tasks but may not provide very good results. The reason is that Lovasz principle aims to learn the global representation of graphs and may not provide discriminative representations for individual nodes.  Here we apply Lovasz principle to GCN and then evaluate the node-level representations in the edge prediction task, comparing with VGAE[2]. We report the AUC scores as follows.
>
> \begin{matrix}
> \hline
>   Method & Cora &  Citeseer &  Pubme   \\\\     \hline
>  VGAE    &  91.4\pm 0.01   & 90.8\pm 0.02  &94.4\pm0.02   \\\\
>  Lovasz  &  84.3\pm 0.24   & 83.1\pm 0.96  &87.2\pm0.37 \\\\
>  \hline
> \end{matrix}
>
> VGAE outperformed the Lovasz principle. Nevertheless, if we convert the node-level tasks to subgraph tasks, i.e., representing each node by a small (local) subgraph containing the node, we may obtain better results for Lovasz principle. We will add more discussion about the application to node-level tasks in the supplementary material.
>
> [2] Kipf T N, Welling M. Variational graph auto-encoders[J].
>
>
> **Question 4: Quality of solver Approximation**
>
> **Response:** Solving for the Lovasz number is a problem with the computation complexity of $O(n^3)$, which can be learned by a GNN with sufficient capacity.
>
> Given a GNN model $\mathcal{F}_W$ trained via the Lovasz principle, the predicted Lovasz number is defined as
>
> $\hat{\vartheta}(G) = \max_ {p \in V} \frac{1}{(\hat{z}^\top \hat{h} _ p)^2}, ~~\text{with} ~~(\hat{z}, \hat{H}) = \mathcal{F} _ W(G)$.
>
> The ground truth Lovasz number of $G$ is denoted as $\vartheta(G)$, which can be computed by the optimization method in [3]. We define an evaluation metric as
>
> $r = \frac{|\hat{\vartheta(G)} - \vartheta(G)|}{\vartheta(G)}$
>
> [3] Wen Z, Yin W. A feasible method for optimization with orthogonality constraints[J].
>
> We use InfoGraph as our GNN $\mathcal{F}_W$ by substituting its InfoMax loss with the Lovasz loss.
> We also propose a constrained optimization method in the **Concern about the regularization approximation instead of exact constraints** part of the rebuttal for Reviewer JpeL [4].
>
> [4]Reviewer JpeL: \url{https://openreview.net/forum?id=0vdEHDwamk&noteId=J2z1RvNm3g}
>
> We select 50 graphs from four dataset and reported the Lovasz number regression rate $r$ for both the regularized ($\mu = 10$) optimization and the constrained optimization for Lovasz principle as follows.
>
> \begin{matrix}
> \hline
>   r& \text{MUTAG} &  \text{PROTEINS} &  \text{DD} & \text{NCI1}  \\\\  \hline
>  regularized\ opt.  &  0.097\pm 0.034   & 0.082\pm 0.021  &0.063\pm 0.011 &0.102\pm 0.036  \\\\
>  constrained\ opt.  &  0.065\pm 0.024   &0.073\pm 0.016   &0.061\pm 0.012 & 0.085\pm 0.023 \\\\
>  \hline
> \end{matrix}
>
> We see that the estimation errors given by $\mathcal{F}_W$ $ are less than 10% in almost all cases. The constrained optimization method [3] performs better than the regularized optimization.
>
> **Question 5: Will the code be publicly available?**
>
> **Response:** Sure, we will make all of our codes publicly available. We will upload the codes to Github. Actually,  our Lovasz principle is a loss that can be used by almost all GNN methods.
>
> **Limitation**
>
> **Response:** We will move more results from the supplementary material to the main paper since NeurIPS allow adding an additional page in an accepted paper.

---

> > ### Comment · Reviewer_2QoV · 2023-08-11
> >
> > I extend my gratitude to the authors for diligently addressing both my concerns and those of the other reviewers. The inclusion of new experiments and observations has strengthened the paper, and I'm convinced it constitutes a valuable contribution worthy of acceptance for publication at NeurIPS.
> >
> > I would urge the authors to incorporate the less favorable results (e.g., VGAE vs. the Lovasz principle) into their Supplementary materials, as a thorough discussion of these aspects could enhance the work's completeness.
> >
> > In light of my concerns being satisfactorily resolved, and the authors' commitment to making their code publicly accessible, I have revised my score from 6 to 7.

---

> > > ### Author Response · Authors · 2023-08-11
> > > **Thanks for the feedback**
> > >
> > > We sincerely thank you for the feedback on our rebuttal and for recognizing our work and raising the score. We will follow your suggestion and make the paper more complete.

---

### Official Review · Reviewer_JWhm · 2023-07-04

**Soundness:** 2 fair
**Presentation:** 3 good
**Contribution:** 3 good
**Rating:** 5
**Confidence:** 4

**Summary:**

This paper presents a technique for unsupervised graph-level (and potentially node-level) representation learning on graphs. The main idea behind the proposed method is the concept of the *Lovász number*, a graph invariant that is related to several graph properties and is computed by solving a min-max optimisation problem on the graph. In particular, to compute this, one needs to optimise a graph-level representation (named the *handle vector*), as well as a set of node-level representations (named the *optimal graph representation*), under certain constraints.

Although the optimisation problem can be solved in polynomial time, it has several disadvantages (it is computationally expensive, it does not have a unique solution, it cannot generalise to unseen graphs, etc.). The authors propose to overcome them by replacing the conventional optimiser with a neural network-based one, i.e. they train a neural network to map each graph to the aforementioned representations, by optimising the Lovasz number objective (regularised by the constraints) with gradient descent. This idea is extended to a more fine-grained setup in order to incorporate subgraph-level information via the so-called Lovasz kernel (a kernel that compares subgraph-level Lovasz numbers). The method is evaluated on several scenarios (unsupervised, semi-supervised and transfer learning), showing consistently competitive performance, and ablated with regards to some algorithmic choices and hyperparameters.


**Strengths:**

**Presentation and motivation**: The paper is generally well-written (with the exception of some clarity issues – see weaknesses). The idea of the Lovasz number is well-explained and illustrated, as well as its role as a graph description is well-motivated. Moreover, the arguments in favour of devising a learning algorithm instead of a conventional optimiser are well-presented.

**Originality and potential impact**. The concept of the Lovasz number is under-explored by the graph ML community (especially from the perspective of graph neural networks), therefore this paper brings a refreshing idea into the field, that might be useful in topics beyond unsupervised learning and pose some interesting new research questions (I mention some of them in the weaknesses section).  It may also incentivise more research exploring other graph properties and combinatorial optimisation problems for unsupervised learning.

**Empirical results**. The method seems to provide consistent performance improvements over multiple unsupervised learning baselines, which indicates its practicality. Moreover, judging from the reported results in the appendix, it seems relatively robust to hyperparameter choices.


**Weaknesses:**

**Lack of theoretical justification & ambiguity regarding what the model is learning**. Given the empirical results, it seems that the method has some appealing property that makes it work well in practice. However, perhaps the biggest weakness of this paper is that it remains unclear what this property is and the authors have not discussed this adequately. In particular,
- It is unclear if the GNN is actually learning the handle vector + optimal graph representations (when optimising Eq. (8)). To be more precise, it is unclear if a GNN *can* express this (mapping a graph to representations that optimise Eq. (2), using a GNN). Given the computational complexity of the problem, I suspect that a GNN (of linear complexity), probably will *not* be able to optimally succeed in this task, but perhaps maybe only approximate it? These are important theoretical questions that should be discussed by the authors.
- Additionally, the constraints are not guaranteed to hold, which casts further doubt on the ability to converge to the desired representations. I think it would be useful to actually test (e.g. for the models whose performance is reported in the tables) how far are the resulting representations from satisfying the constraints, and how good are they in approximating the Lovasz number. Moreover, do the node features (that are taken into account by the GNN, but not in Eq. (2)) affect the resulting representations? Reporting these results will provide further insights into what these models are actually learning.
- I am wondering if it is actually desired to learn to infer the handle vector since the results in Table 1 (reported as Lovasz number) are significantly worse than the proposed method (reported as Lovasz principle). Isn’t this contradictory to the initial motivation?
- Eventually, I am wondering why is this method so good (and better than the Lovasz number) at solving the tested downstream tasks. What kind of information do the resulting representations contain? Maybe something related to substructures? Note that the arguments mentioned by the authors are attractive properties of the Lovasz number, but not the handle vector.


**Clarity issues and inadequate justification of design choices**. Some of the choices made by the authors are not sufficiently justified and I’d like to encourage them to provide more details. In particular,
- Section 3, Eq. (8):  Wouldn’t it be possible to guarantee the satisfaction of the 2nd constraint (unit-length constraint for the handle vector) by construction? For example, isn’t it possible to extract an unconstrained graph representation using a GNN and then divide it by its norm? It would be interesting to discuss if something similar could happen for the 1st constraint.
- Section 3, Eq. (10): Similarly, here why does one need a second encoder (and therefore the third regularisation term)? Why not use the same encoder and compute both objectives with that?
- Section 4. This section lacks the required level of detail and some parts are inadequately explained.
  - I am a bit confused by the iterative nature of the algorithm here. Could the authors elaborate? Do the iterations refer to gradient iterations, i.e. is this akin to bi-level optimisation? If this is the case, why is the first term of Eq. (14) not indexed by $t$?
  - How many subgraphs are needed to obtain a good estimate of the kernel?
  - The role of the spectral embedding idea is unclear to me. Why are the handle vectors encouraged to be orthogonal in Eq. (13), and most importantly what is the role of the last term?



**Questions:**

**Experiments**.
- Could the authors discuss how the loss terms behave in practice (for both Eq. (8) and Eq. (13))? Does one term dominate over the others, and how far are the constraints from being satisfied (this is also related to my first concern)?
- More experimental/implementation details are needed regarding the transfer learning experiment. The authors mentioned that these can be found in the supplementary material, but I couldn't find enough details. For example, it was unclear how the pre-training is performed and where the Lovasz number loss is used.
- Appendix Section 3.2.: I found it a bit puzzling that the performance does not deteriorate when $\mu$ is too large (e.g. 1000). I would expect that these values would imply an unsuccessful optimisation of the first term of the objective. Have the authors tested this (e.g. by testing if the learned graph and node-based representations lead to a value close to the Lovasz number)?
- Appendix section 4.1. is a bit confusing. It seems that in several cases, using the orthonormality constraint deteriorates performance. Could the authors comment on this?
- Did the authors reimplement the baselines, in order to ensure fairness of comparison?


**Minor**:
- If I am not mistaken the Lovasz number is computed by also optimising over the dimension of the handle vector. Could the authors account for this as well?
- Could the authors provide more details on the computational complexity of computing the Lovasz number (this is relevant to the question if a GNN can solve this problem to optimality)?
- In the last two sections of Table 1, I assume that the authors simply used the neural network architectures of the mentioned papers, however the way the results are presented is a bit confusing. Maybe explicitly mentioning it in the caption would help. Moreover, the important element that should be highlighted in this table, is if the Lovasz optimisation objective improves against the baseline one, so maybe the authors would like to reorder the rows in the table to emphasise this comparison.
- It might be also useful to add some results in Table 1 from end-to-end supervised learning methods, in order to obtain a more complete picture of the capabilities of the unsupervised methods.
- Have the authors considered using the node-level representations for node-wise tasks?
- L291: “For those contrastive…naïve strategy”. Do the authors refer to the baselines? This should be clarified to avoid confusion.
- Typo: Section 4.3.





**Limitations:**

The authors make a short discussion on the limitations in the conclusion, but I think more should be mentioned (especially regarding the ambiguity regarding the learned representations - see the weaknesses section).  No foreseeable negative societal impact.

---

> ### Author Rebuttal · Authors · 2023-08-08
>
> **We highly appreciate your comprehensive review, insightful comments, and positive assessment.**
>
>
> **Weakness: Lack of theoretical justification & ambiguity regarding what the model is learning**
>
> * We assumed that the training data ($N$ graphs) are from the same distribution. Thus, the graphs should share some common structures or properties. For instance, when two graphs are in the same class, their orthonormal representations should be similar and the Lovasz numbers of them or their subgraphs are similar [Johansson et al., 2014].  Therefore, it is expected that a sufficiently large neural network is able to learn the intrinsic structure from the training graphs and then solve the optimization related to the orthonormal representation and handle vector in an end-to-end manner. The idea is similar to the general idea of learning to optimize [Li and Malik ICLR 2017; Chen et al. JMLR 2022]. Please also refer to our response to Q4 of Reviewer 2QoV.
>
> * Although in our paper we only reported the results given by the regularized optimization, we actually have found that the performances are similar to the constrained optimization (solved by projected update or exact penalty method). Moreover, in the ablation study of the supplementary material, we can see that the method is not very sensitive to the penalty parameter $\mu$. This is due to the good quality of the handle vector, though not the exact solution. In addition, the node features are indeed useful though  Eq. (2) is not related to the node features.
>
> We define the following metric:
> $r = \frac{|\hat{\vartheta(G)} - \vartheta(G)|}{\vartheta(G)}$
>
> We have the following results (see our response to Reviewer 2QoV). The estimation errors are less than 10% in almost all cases.
> \begin{matrix}
> \hline
>   r& \text{MUTAG} &  \text{PROTEINS} &  \text{DD} & \text{NCI1}  \\\\  \hline
>  regularized\ opt.  &  0.097\pm 0.034   & 0.082\pm 0.021  &0.063\pm 0.011 &0.102\pm 0.036  \\\\
>  constrained\ opt.  &  0.065\pm 0.024   &0.073\pm 0.016   &0.061\pm 0.012 & 0.085\pm 0.023 \\\\
>  \hline
> \end{matrix}
>
> * The bad performance of the Lovasz number in Table 1 stems from the fact that the orthonormal representation and handle vector for a graph is not unique, which was explained in lines 149-157 in the main paper. Thus, even if two graphs are exactly the same, the solver for (2) may provide two very different handle vectors, which leads to bad performance in downstream tasks.
>
> * In our Lovasz principle, because of using neural network and the end-to-end formulation, the orthogonal representation and handle vector  of each graph are unique and rotation-invariant. That's why it is much better than the Lovasz number in Table 1. By the way, in our Lovasz principle, the end-to-end formulation learned the distribution information of the $N$ graphs successfully, while solving the Lovasz number for each graph by a solver independently failed to learn the distribution information.
>
>
> **Clarity issues and inadequate justification...**
>
> * Yes. We have provided a solution in the response to Reviewer JpeL. Besides this projection-based strategy, we can use the exact penalty method, i.e., increasing $\mu$ gradually, which is a common strategy for constrained optimization.
>
> * The InfoGraph method used two encoders (see Eq. (17)) for the semi-supervised learning and the works (GraphCL, AD-GCL, JOAO, AutoGCL, etc.) followed the convention.  To ensure a fair comparison, we considered two encoders in our work.
>
> * -----
>    * The optimization related to (14) is not a bi-level optimization. In each iteration $t$, the objective function is (14) and the parameters are updated by a mini-batch optimization. We can just regard the $K_{ij}^{(t-1)}$ in (13) as a constant not related to the network parameters, so the optimization problem changes in every iteration. The idea is very similar to that of the iteratively reweighted least squares. We will provide more explanation in the revision.
>    * We follow the implementation code of the Lovasz kernel [Joh+14] in the Grakel framework [1]. When dealing with large-size graphs, Grakel [1] selects subgraphs of sizes 3, 4, and 5, while excluding others. This is their code implementation for achieving a well-approximated Lovasz kernel using samples as in [Joh+14, Theorem 1].--------[1] Grakel \url{https://ysig.github.io/GraKeL/0.1a8/kernels/lovasz_theta.html}
>    * The orthogonality of the handle vectors and the last term (sum to 1) are constraints of spectral embedding [2,3].
> [2] Belkin M, Niyogi P. Laplacian eigenmaps for dimensionality reduction and data representation[J].
> [3] \url{https://perso.telecom-paristech.fr/bonald/documents/spectral.pdf}
>
>
> **Experiments**
>
> * The loss behavior is reported in the attached PDF file.
>
> * We will provide more details about the experimental setting of transfer learning.
>
> * The performance is indeed not sensitive to $\mu$ if it is in the range of $[0.1,1000]$. When $\mu$ is too large, the learned Lovasz number is quite different from the one given by a solver.
>
> * Yes, in some cases, the orthonormality constrainn reduced the accuracy. One possible reason is that the baseline method on a specific dataset learned diverse features which lead to difficulties in making them orthonormal.
>
> * We reproduced the results of the baselines and found that they are very close to the results reported in the original paper. Therefore, we just report the original results of the authors in the tables. This convention has been followed by previous works such as  [Hu et al., 2019; You et al., 2021; Yin et al., 2022]. It should be pointed that for our methods, based on the codes released by [You et al., 2021; Yin et al., 2022], we just replaced the Infomax principle by our Lovasz principle, without changing any structure or optimization parameter in the original codes. Therefore, the comparison is fair.
>
> **Minor issues**
>
> We thank the reviewer's detailed comments again and we will address these minor issues in the revised paper.

---

> > ### Comment · Reviewer_JWhm · 2023-08-20
> > **Response to Authors**
> >
> > Dear Authors,
> >
> > Thank you for your reply. Although some of my questions have been answered, my major concern regarding what the GNN actually learns and if vanilla GNNs can express the Lovasz number remains.
> >
> > - An intuitive explanation about why the method works so well is still missing. Why should one choose to learn to solve for the Lovasz number and not for a different graph property? The lack of sensitivity to $\mu$ makes me even more sceptical.
> > - In their response to Reviewer 2QoV, the authors claim that “Solving for the Lovasz number […] can be learned by a GNN with sufficient capacity”. However, this is not proven but only tested experimentally, which in fact indicates that the Lovasz number can be approximated, not optimally computed. I am still sceptical if a GNN can indeed express the solution to this problem, mainly due to its computational complexity, as I already mentioned in my initial review; note that many graph properties are proven to be incomputable by vanilla GNNs. In case this is also true here, one possible proof idea is to find a pair of graphs (counterexample) that have the same Lovasz number but are indistinguishable by vanilla GNNs.
> >
> > *Minor*. Additionally, the motivation behind adapting the spectral embedding idea is still not adequately explained.
> >
> > For the time being, I will keep my score unchanged due to the above reservations.

---

> > > ### Author Response · Authors · 2023-08-21
> > > **Further clarification**
> > >
> > > Dear Reviewer,
> > >
> > > We thank you for your further comments. Our responses are as follows.
> > >
> > > 1. The good performance is owing to the good property of the handle vector of Lovasz number. According to Figure 1 (Pentagon example for Lovász number, also presented in the first figure of the PDF in the global rebuttal) in our paper,  intuitively, the handle vector can be viewed as the handle of an umbrella, where the ribs are nodes of the graph. Thus, the handle vector is able to capture the global structure of a graph and becomes a natural vector representation of the graph.
> > > Importantly, according to the definition of Lovasz number, i.e.,
> > > $$\qquad\vartheta(G):=\min _{\boldsymbol{c}, \boldsymbol{U} \in \mathcal{U}} \max _{p \in V} \frac{1}{\left(\boldsymbol{c}^{\top} \boldsymbol{u}_p\right)^2}$$ we see that the handle vector $\mathbf{c}$ is the vector with smallest angles to the most spreading orthonormal representation of a graph. So the handle $\mathbf{c}$ can be regarded as a 'centroid' of ribs of the umbrella. The computation of $\mathbf{c}$ is based on the representations of all nodes in a graph.
> > > We actually have studied other graph properties but did not find such a vector. It is worth noting that Lovasz number is polynomial-time solvable (e.g. SDP) while most other graph properties such as the clique number are NP-hard.
> > >
> > > 2. The Lovasz number is polynomial-time computable and can be solved by SDP [Grötschel et al.,1981; Galli and Letchford, 2017]. Currently, we cannot prove that the Lovasz number can be exactly solved by a GNN. But we think it is an important problem to be studied in the future and it is difficult to solve all problems in one paper. The are many successful papers on learning to optimize. They use neural networks to solve convex $\ell_1$-minimization [1], neural network training [2], black-box optimization [3], combinatorial optimization [4], etc., but they **did not prove** that the corresponding problems can be exactly solved by the neural networks.
> > >
> > > [1] Karol Gregor and Yann LeCun. Learning fast approximations of sparse coding. In Proceedings of the 27th international conference on international conference on machine learning, pages 399–406, 2010.
> > >
> > > [2] Marcin Andrychowicz, Misha Denil, Sergio Gomez, Matthew W Hoffman, David Pfau, Tom Schaul, Brendan Shillingford, and Nando De Freitas. Learning to learn by gradient descent by gradient descent. In Advances in neural information processing systems, pages 3981–3989, 2016.
> > >
> > > [3] Yutian Chen, Matthew W Hoffman, Sergio G´omez Colmenarejo, Misha Denil, Timothy P Lillicrap, Matt Botvinick, and Nando Freitas. Learning to learn without gradient descent by gradient descent. In International Conference on Machine Learning, pages 748–756, 2017.
> > >
> > > [4] Elias Khalil, Hanjun Dai, Yuyu Zhang, Bistra Dilkina, and Le Song. Learning combinatorial optimization algorithms over graphs. Advances in neural information processing systems, 30:6348–6358, 2017.
> > >
> > > In our response to Reviewer 2QoV, we used experiments to show that the solutions given by our neural network $\mathcal{F}_W$ are close to the solutions given by an exact solver. Given a GNN model $\mathcal{F}_W$ trained via our Lovasz principle, the predicted Lovasz number is defined as
> > > $$
> > > \qquad\hat{\vartheta}(G)=\max _{p \in V} \frac{1}{(\hat{z}^\top \hat{h} p)^2}, \text { with }(\hat{z}, \hat{H})=\mathcal{F}_W(G)
> > > $$
> > > The ground truth Lovasz number of $G$ is denoted as $\vartheta^\ast(G)$, which is computed by the solver exactly. We define an evaluation metric as $$\qquad r=\frac{|\hat{\vartheta}(G)-\vartheta^\ast(G)|}{\vartheta^\ast(G)},$$
> > > the smaller the better.
> > >
> > > We use InfoGraph as our GNN $\mathcal{F}_ W$ by substituting its InfoMax loss with the Lovasz loss. We also consider a constrained optimization to train $\mathcal{F}_ {W}$.
> > >
> > > We select 50 graphs from four datasets and reported the Lovasz number regression rate $r$ for both the regularized $(\mu=10)$ optimization and the constrained optimization for Lovasz principle as follows.
> > >
> > > **Table: Approximation Error for the exact Lovasz number**
> > > \begin{matrix}
> > > \hline
> > >  & MUTAG & PROTEINS & DD & NCI1 \\\\
> > > \hline regularized opt. & 0.097 \pm 0.034 & 0.082 \pm 0.021 & 0.063 \pm 0.011 & 0.102 \pm 0.036 \\\\
> > > constrained opt. & 0.065 \pm 0.024 & 0.073 \pm 0.016 & 0.061 \pm 0.012$ & 0.085 \pm 0.023 \\\\
> > > \hline
> > > \end{matrix}
> > > We see that the estimation errors given by $\mathcal{F}_W $ (regularized or constrained) are **less than 10%** in almost all cases. This demonstrates that the solutions provided by a neural network $\mathcal{F}_W$ are quite good compared to the exact solutions given a solver. By the way, in our response to Reviewer 2QoV, we showed that the handle vectors are discriminative for graphs from different classes.
> > >
> > > We hope this clarification could alleviate your concerns and we are still considering how to prove that the neural networks are able or unable to solve the Lovasz number exactly, though it is very challenging. Thank you again.
> > >
> > > Sincerely,
> > >
> > > Authors

---

> > > > ### Author Response · Authors · 2023-08-21
> > > > **Explanation for adapting the spectral embedding and sensitivity to $\mu$**
> > > >
> > > > **Spectral embedding**
> > > > As explained in lines 209-212 of our paper, our Lovász principle does not explicitly utilize the Lovász number in graph embedding, though the Lovász numbers of subgraphs can be useful in comparing graphs [Johansson et al., 2014]. Therefore, we proposed to use subgraph Lovász number to enhance Lovász principle based graph representation learning. We take advantage of the Lovász-ϑ kernel proposed by [Johansson et al., 2014] to regularize the representation learning:  If two graphs $G_ i$ and $G_ j$ are similar
> > > > in terms of the Lovász-ϑ kernel (i.e., $K_ {ij}$ is large), their graph-level representations $\mathbf{z}_ i$ and $\mathbf{z}_ j$ should be close to each other. Inspired by the spectral embedding [Belkin2001], we obtain:
> > > > \begin{equation}
> > > >     \begin{aligned}
> > > >          \min_ {Z} \sum_ {i= 1}^N \sum_{j= 1}^N K_ {ij} ||{\bf z_ i} - {\bf z_ j} ||_ 2^2, ~~~
> > > >         \text{s.t.} ~~ Z^\top Z = I_ d, Z^\top \mathbf{1} = \mathbf{0}.
> > > >     \end{aligned}
> > > > \end{equation}
> > > >
> > > > The reasons why we use such constraints are as follows.
> > > >
> > > >  * In spectral embedding, the solution is not unique. Given an arbitrary constant vector $\mathbf{c}$, if $Z$ is one of the solution of spectral embedding, then $\hat{Z} := \{\mathbf{z_ i} + \mathbf{c}, \text{for all} i = 1, ..., n\}$ is also a solution, because  $||(\mathbf{z_i} + \mathbf{c}) - (\mathbf{z_j} + \mathbf{c})  ||_2^2 = ||{\mathbf{z}_i} - {\mathbf{ z}_j} ||_2^2$. Thus they introduce the constraint $Z^\top \mathbf{1} = \mathbf{0}$ to ensure all the embeddings are located near the original point.
> > > >
> > > >  * With only the constraint $Z^\top \mathbf{1} = \mathbf{0}$, the solution for spectral embedding is trivial, $Z = 0$ is one of the solutions for spectral embedding. This trivial solution reveals nothing about the structural information of a graph, thus they introduce another constraint $Z^\top Z = I_d$ such that the solution is non-trivial and strongly related to the eigenvectors of the matrix $K$.
> > > >
> > > > The constrained optimization is relaxed as the following regularized optimization:
> > > >
> > > > $$\qquad\mathcal{L}_ {\mathrm{SLN}}^{(t)}:=\sum_ {i=1}^{|\mathcal{G}|} \sum_ {j=1}^{|\mathcal{G}|} K_ {i j}^{(t-1)}\left\Vert\boldsymbol{z}_ i^\phi-\boldsymbol{z}_ j^\phi\right\Vert_ 2^2+\gamma\left(\left\Vert\boldsymbol{Z}_ \phi^{\top} \boldsymbol{Z}_ \phi-\boldsymbol{I}_ d\right\Vert_ F^2+\left\Vert\boldsymbol{Z}_ \phi^{\top} \mathbf{1}_ {N \times 1}\right\Vert_ 2^2\right)$$
> > > >
> > > > and the overall loss is $\mathcal{L}_ {\mathrm{ELo}}^{(t)}:=\mathcal{L}_ { \mathrm{Lo}}+\eta \mathcal{L}_ {\mathrm{SLN}}^{(t)}$. The idea of the optimization is similar to iteratively reweighted least square. For interaction $t$, we compute $K_ {i j}^{(t-1)}$ using
> > > > $$\qquad K_{i j}^{(t-1)}=\sum_ {S_i \subset V_ i} \sum_{S_ i \subset V_ i} \frac{\delta\left(\left|S_ i\right|,\left|S_ j\right|\right)}{C_ {S_ i, S_ j}} k\left(\vartheta_{S_ i}^{(t-1)}\left(G_ i\right), \vartheta_ {S_ j}^{(t-1)}\left(G_ j\right)\right)$$
> > > > where $\vartheta_ {S_ i}^{(t-1)}\left(G_ i\right)=\max _ {p \in S_ i} \frac{1}{\left(\boldsymbol{z}_ i^{(t-1)^{\top}} \boldsymbol{u}_ p^{(t-1)}\right)^2}$ is conducted based on the network learned at iteration $t-1$. Then we perform gradient descent for $\mathcal{L}_ {\mathrm{ELo}}^{(t)}$ with $K_ {i j}^{(t-1)}$ fixed as constants.
> > > >
> > > > **Sensitivity to $\mu$**
> > > >
> > > > Indeed, in many cases, the classification (SVM) performances are not sensitive to $\mu$. One possible reason is that when $\mu$ is in a reasonable range, i.e., $[1,1000]$, SVM is robust to $\mathbf{z}_i$ given by different $\mu$. This, on the other hand, demonstrated the stability of our methods. Note that $\mu$ actually can influence the results significantly. For instance, when $\mu=10^5$, the performance degraded quickly.
> > > >
> > > > **Thank you very much for spending time on our rebuttal**.

---

### Official Review · Reviewer_4DKt · 2023-07-05

**Soundness:** 3 good
**Presentation:** 3 good
**Contribution:** 3 good
**Rating:** 7
**Confidence:** 4

**Summary:**

This paper centers on graph-level representation learning, aimed at converting graphs into vectors useful for downstream tasks like graph classification. The authors propose a unique learning principle named the Lovász principle, inspired by the Lovász number in graph theory. The Lovász number, a real number serving as an upper bound for a graph's Shannon capacity, has strong ties to various global graph characteristics. The authors suggest that the handle vector, used for calculating the Lovász number, could be an effective choice for graph representation given its ability to capture global graph properties. However, its direct application poses challenges. To address these, the authors propose using neural networks to offer the Lovász principle. They also present an enhanced Lovász principle capable of directly and efficiently utilizing subgraph Lovász numbers. Experimental results demonstrate competitive performance of these Lovász principles in comparison to the baselines in both unsupervised and semi-supervised graph-level representation learning tasks.

**Strengths:**

1. The authors' proposal of Lovász theta kernels for graph representation learning shows great promise, thanks to its theoretical foundation.
2. The paper includes thorough experiments conducted in various settings, including unsupervised, semi-supervised, and transfer learning. These experiments provide strong evidence of the effectiveness of the proposed methods.

**Weaknesses:**

1. Although the author proposed a new graph kernel, the contribution seems slightly small to me.
2. The author did not demonstrate the framework of the model.
3. Some classic baselines were not compared.

**Questions:**

1. I strongly suggest that the author discuss the application of graph kernel in updated scenarios, such as graph prompt learning. It is highly recommended that the authors consider citing the prompt learning on graph[1,2] paper and explore its application in conjunction with the proposed method in this paper, especially for unsupervised settings. This would enhance the discussion and potential application of prompt-based models.

 [1] Graphprompt: Unifying pre-training and downstream tasks for graph neural networks. Z Liu, X Yu, Y Fang, X Zhang - Proceedings of the ACM Web Conference 2023, 2023

 [2] Sun, M., Zhou, K., He, X., Wang, Y. and Wang, X., 2022, August. Gppt: Graph pre-training and prompt tuning to generalize graph neural networks. In Proceedings of the 28th ACM SIGKDD Conference on Knowledge Discovery and Data Mining.

2. Providing a framework of the proposed model can indeed enhance the clarity of its functionality.
3. Why semi-supervised learning baseliens such as GCN, GAT, GIN are not compared in the paper?

**Limitations:**

See weaknesses.

---

> ### Author Rebuttal · Authors · 2023-08-08
>
> **Question 1: Applications for graph prompt learning**
>
> **Response:** We appreciate your suggestion and will cite papers [1,2] in the revised paper. Graph prompt learning involves transferring learning in graphs. Challenges arise when refining a pre-trained Graph Neural Network (GNN) for specific tasks using limited labeled data. Fortunately, when the GNN is initially pre-trained based on the Lovász principle, it acquires representations geared towards solving the Lovász number problem, rather than being tailored to a specific task. This pre-trained GNN, rooted in the Lovász principle, can seamlessly adapt to downstream tasks' unlabeled data to learn their orthogonal representations. Moreover, we can determine the true Lovász number of the downstream task's data and employ the pre-trained GNN to predict the Lovász number for the same data. The regression loss between the true Lovász number and the predicted Lovász number serves as a prompt to fine-tune the pre-trained GNN. The Lovász principle, being a graph-based learning principle, is exceptionally well-suited for graph prompt learning.
>
> **Question 2: Lack of model framework**
>
> **Response:** We in this rebuttal provide a model framework in the attached PDF file.
> Lovasz principle Eqn. (8) is actually a loss function Eqn. (9). This loss function finds utility in all graph learning models satisfying Eqn. (7), including examples like InfoGraph [3] and GraphCL [4]. To illustrate, consider InfoGraph [3]: by substituting the original InfoMax loss with the loss function from Eqn. (9), we establish an InfoGraph framework guided by the Lovasz principle. Importantly, the Lovasz principle's applicability isn't confined to any specific GNN model or graph data; rather, it offers a versatile approach across various contexts. Its generality is parallel to that of the InfoMax principle.
>
> [3]Fan-Yun Sun, Jordan Hoffmann, Vikas Verma, and Jian Tang. Infograph: Unsupervised and semi-
> 476 supervised graph-level representation learning via mutual information maximization.
>
> [4]Yuning You, Tianlong Chen, Yongduo Sui, Ting Chen, Zhangyang Wang, and Yang Shen. Graph
> 527 contrastive learning with augmentations.
>
>
> **Question 3: Comparison with  some basic GNN models**
>
> **Response:** When it comes to learning graph-level representations, our primary comparison baselines (InfoGraph[3], GraphCL[4], AD-GCL[5], JOAO[6], AutoGCL[7]) stand out as the most current and influential methods spanning from 2019 to 2022, each boasting high citations on Google Scholar. Notably, these approaches are considerably more sophisticated and effective compared to fundamental GNN models like GCN, GAT, and GIN. Furthermore, these updated methods incorporate basic GNNs as foundational elements, with GIN serving as a fundamental component in all five techniques. Particularly, InfoGraph employs five GINs to acquire $H$ and  $z$. It's worth mentioning that graph-level representation learning achieved by merely adding a READOUT function [8] to basic GNNs doesn't lend itself well to direct comparison.
>
>
> [5] Susheel Suresh, Pan Li, Cong Hao, and Jennifer Neville. Adversarial graph augmentation to improve
>  graph contrastive learning.
>
> [6] Yuning You, Tianlong Chen, Yang Shen, and Zhangyang Wang. Graph contrastive learning automated.
>
> [7] Yihang Yin, Qingzhong Wang, Siyu Huang, Haoyi Xiong, and Xiang Zhang. Autogcl: Automated graph contrastive learning via learnable view generators.
>
> [8] Tian Xie and Jeffrey C Grossman. Crystal graph convolutional neural networks for an accurate and
> interpretable prediction of material properties.
>
> **We sincerely thank the reviewer for recognizing the contribution of our work.**

---

> > ### Comment · Reviewer_4DKt · 2023-08-20
> >
> > Thank you for your comprehensive response. It has thoroughly addressed my queries. Consequently, I've adjusted my score to 7.

---

> > > ### Author Response · Authors · 2023-08-20
> > > **Authors‘ feedback**
> > >
> > > Dear Reviewer 4DKt,
> > >
> > > We thank you so much for your constructive comments and for adjusting your score.
> > >
> > > Sincerely,
> > >
> > > Authors

---

### Official Review · Reviewer_1BFQ · 2023-07-07

**Soundness:** 3 good
**Presentation:** 2 fair
**Contribution:** 2 fair
**Rating:** 6
**Confidence:** 3

**Summary:**

The authors present a new method for graph representation in supervised, semi-supervised and transfer learning based on the Lovasz theta function [Lov79]. They also incorporate local information using Lovasz subgraph numbers inspired by the work on Lovasz theta kernal [Joh+14]. They also present empirical evaluation and show that their method is competitive with state-of-the-art.


**Strengths:**

- The paper is easy to follow for the most part.
- The empirical evaluation shows the method performs well compared to existing methods.


**Weaknesses:**

Some rewriting needed on Section 4 and minor typos. Also ablation studies and final parameter selection still a bit unclear.

**SLN**

- Lines 209-211. Please define "subgraph Lovasz number" explicitly.

- Lines 214-218. The description of Lovasz $\vartheta$ kernel [Joh+14] is misleading.
    - Lines 214-216. Possibly I misunderstood this, but subgraph lovasz value as defined in [Joh+14, Definition 2] can be well-approximated using $O(n \log n)$ samples [Joh+14, Thm. 1].

    - Lines 217-218. In so far as the kernel being a pair-wise method and not capturing the "global structure" of $\cal G$, here is the trivial equivalent using the kernel:  $ \sum_{i=1}^{\cal G} \sum_{j=1}^{\cal G} \hat{k}_{\tt Lo}(G_i, G_j) $.

- Lines 226-229. How is $K^{(t)}_{ij}$ computed, more precisely the number of  sampled subgraphs and how those are selected?

**Ablation and default parameters**

- The ablation study has explored parameter choices for $\mu$, $\eta$, etc. over log-ranges for the different graph datasets. However, what is not so clear is the influence of graph dataset properties (e.g., size of the graphs, density, node properties, etc.)

- It would be great if the authors could present preferably in the main text or the appendix, a set of sane starting points. Of course, for each use-case, HPO should be done but often, the practitioner take is to go with defaults initially.

**Minor typos**

- Appendix. Section 4. lines 80-91 "orthnormal" -> "orthonormal"
- Appendix. Section 5. line 95. "guarantee" -> "guaranteed"



**Questions:**


- It would be interesting to see in eqn (9) how the different loss terms during training corresponding to the theta-value ($l1$) and  orthonormal representations ($l2$ and $l3$) and similarly for SLN when thinking of the final READOUT as non-linear analogues of the Lovasz representation.

---

> ### Author Rebuttal · Authors · 2023-08-08
>
> **We sincerely thank the reviewer for recognizing our contribution. Our responses are as follows.**
>
> **On SLN**
>
> * The definition of "subgraph Lovasz number" is in [Joh+14, Definition 2]. Given a graph $G = (V, E)$, let $G[S]$ be a subgraph of $G$ induced by a vertex subset $S \in V$. The "subgraph Lovász number" is defined as
> $ \vartheta_S (G) = \min_c \max_{u_i \in U_{G|B}} \frac{1}{(c^\top u_i)^2}, $
> where $U_{G|S} := \{u_i \in U_G | i \in S\}$ and $U_G$ is the orthonormal representations associated with $\vartheta(G)$. Note that in general $\vartheta_S (G) \neq \vartheta(G[S])$. We added this definition to the revised paper.
>
> * You're right. In the case of larger graphs, [Joh+14] doesn't calculate the Lovasz kernel for all graphs; instead, they employ a sampling strategy to decrease the run-time. Similarly, in our experiments involving large graphs, we also utilize sampling techniques to enhance run-time efficiency. The equivalent totally sum-up kernel you mentioned is actually effective in capturing the graph information across the entire graph dataset $\mathcal{G}$. We will modify the formulation accordingly.
>
> * We follow the implementation code of the Lovasz kernel [Joh+14] in the Grakel framework [1]. When dealing with large-size graphs, Grakel [1] selects subgraphs of sizes 3, 4, and 5, while excluding others. This is their code implementation for achieving a well-approximated Lovasz kernel using samples as in [Joh+14, Theorem 1].
>
> [1] Grakel \url{https://ysig.github.io/GraKeL/0.1a8/kernels/lovasz_theta.html}
>
> **On ablation study and default parameters**
>
> * Thanks for pointing it out. We will add a comprehensive description of the data statistics, parameter settings, and starting points for each dataset in the supplementary material. We will also release our codes publicly.
>
> **Question: Analysis of Losses and the non-linear READOUT analogues**
>
> * The role of each loss has been analyzed in the supplementary material. In Appendix 3 (Parameter sensitivity analysis), we choose the weight of each loss from $10^{-3}$ to $10^6$. The performance of very small hyperparameters can be regarded as the ablation for each loss.  The Ablation study in Appendix 4  also indicates that orthonormal regularization is of great importance in our Lovasz principle.
> by the way, instead of using a fixed $\mu$, we can increase the value of $\mu$ gradually, which corresponds to the exact penalty method and is able to well approximate the constrained optimization.
>
> * The role of Lovasz principle is very similar to the READOUT function or pooling layers in graph-level representation learning. The typical READOUT [2] methods tend to be straightforward, like average pooling or max pooling. The Lovasz principle, on the other hand, is firmly rooted in graph theory, stemming from its connection to the Lovasz number.
>
> [2] William L Hamilton. Graph representation learning. Synthesis Lectures on Artifical Intelligence and Machine Learning
>
> **We're grateful for your suggestion. We plan to enhance the clarity of the definition and presentation of SLN in the main text, and we'll also include additional experimental details about the data and default parameters in the supplementary materials.**

---

> > ### Comment · Reviewer_1BFQ · 2023-08-20
> >
> > Thank you for addressing my concerns which have been satisfactorily addressed.

---

> > > ### Author Response · Authors · 2023-08-21
> > > **Authors' comment**
> > >
> > > We highly appreciate your feedback and your recognition of our work.

---

### Official Review · Reviewer_JpeL · 2023-07-07

**Soundness:** 3 good
**Presentation:** 3 good
**Contribution:** 2 fair
**Rating:** 3
**Confidence:** 4

**Summary:**

This submission presents a new approach to graph-level representation learning inspired by the Lovász number in graph theory. Specifically, the study proposes using the Lovász principle as a novel framework for unsupervised/semi-supervised graph representation learning. It offers a method to utilize handle vectors, which capture a graph's global features through neural networks. An enhanced Lovász principle is also proposed, which efficiently uses subgraph Lovász numbers, thereby ensuring similar graphs have similar representations. Experimental results show that the Lovász principles outperform other graph representation methods (such as InfoMax) in unsupervised learning, semi-supervised learning, and transfer learning. Therefore, the Lovász principles offer a promising new approach to graph-level representation learning.

**Strengths:**

There are three main strengths:

1. A new approach based on the Lovász principle is proposed for graph-level representation learning. This principle, inspired by the Lovász number in graph theory, adds a new perspective and strategy to graph learning, shifting from traditional methods.

2. Some subgraph tricks are used to enhance the performance of Loasz-based GNN models.

3. The Lovász principles outperform other methods in experiments on unsupervised learning, semi-supervised learning, and transfer learning.


**Weaknesses:**

Weaknesses:

1. Compared with current GNN models, such as Infomax-based and kernel-based methods, there are indeed some gains in terms of ACC on the task of graph classification. However, the overall gained performance in Table 1,2, and 3 is not very significant, especially when you consider the variances. This may question whether Lovász Principle can provide more useful global information than InfoMax. The other concern is that it is unclear how the approximation affects the whole performance since the final vectors used in the proposed models are approximated.

2. In terms of the novelty of using the Lovász Principle for designing GNN models, one of my main concerns is that this idea is not very novel. There is a lack of discussion on the difference between current work and previous works. See related works in [1]. In terms of run-time comparison, there is no significant reduction compared with InfoMax based on the run-time table in the appendix.

[1] Yadav, P., Nimishakavi, M., Yadati, N., Vashishth, S., Rajkumar, A. and Talukdar, P., 2019, April. Lovasz convolutional networks. In The 22nd international conference on artificial intelligence and statistics (pp. 1978-1987). PMLR.



**Questions:**

Overall, this paper has a good idea of using the Lovasz Principle (the handle vector) to present the graph and incorporate this idea to design GNN models. However, there is some related work, and the authors should have discussed it in the submission.

Q1: How could you deal with large-size graphs? For example, the datasets RDT-B and RDT-M5K were used in [2]. My concern is that it is unclear how you can estimate C_{S_i,S_j} accurately when n, the number of nodes in the graph, is large.

Q2. Compared with InfoMax, why the performance of InfoMax-based (in terms of best ones) are very close to the proposed? More discussions are needed.

[2] Sun, Fan-Yun, Jordan Hoffmann, Vikas Verma, and Jian Tang. "Infograph: Unsupervised and semi-supervised graph-level representation learning via mutual information maximization." arXiv preprint arXiv:1908.01000 (2019).

Some minors:

1. It is helpful to list out all dataset statistics for people who are unfamiliar with this area.



**Limitations:**

None.

---

> ### Author Rebuttal · Authors · 2023-08-06
>
> **Concern about the novelty**
>
> **Response:** We have to clarify that our method is **very different** from [1] (Lovasz Convolutional Networks (LCN)). LCN was motivated by the observation that removing certain vertices from a graph doesn't affect the graph's global properties such as the Lovasz number. LCN replaces the affinity matrix $\hat{A}$ of classical GCN by a Lovasz kernel $K = \frac{A}{-\lambda_{\text{min}}(A)} + I$ and hence yields the following model: $f({X}, {K}) = \text{softmax}({K} \text{ReLU}({K}{X}{W}^{(0)}){W}^{(1)})$. In contrast, our Lovasz principle is inspired by the umbrella analogy in Figure 1. We summarize the differences as follows:
> * The goal of LCN is node embedding and classification while the goal of our Lovasz principle is graph representation learning.
> * More importantly, according to $f({X}, {K})$, we see that LCN does not involve any optimization related to Lovasz number. It just replaced $\hat{A}$ of classical GCN with a predefined $K$.  In contrast, our Lovasz principle transcends specific GNN designs and it solves the optimization of Lovasz number and orthogonal representation via deep neural networks:
> $\mathcal{L}_ {\mathrm{Lo}}=\sum_{i=1}^{|\mathcal{G}|} \max _{p \in V_i} \frac{1}{\left(\left(\boldsymbol{z}_i^\phi\right)^{\top} \boldsymbol{h}_p^\theta\right)^2}+\mu\left(\left\Vert\boldsymbol{M}_i \odot\left(\boldsymbol{H}_i^\theta\left(\boldsymbol{H}_i^\theta\right)^{\top}-\boldsymbol{I}_n\right)\right\Vert_F^2+\left(\left(\boldsymbol{z}_i^\phi\right)^{\top} \boldsymbol{z}_i^\phi-1\right)^2\right)$
> * Our Lovasz principle yields better graph representations than other methods such as the Infomax principle.
>
> We sincerely thank the reviewer for pointing out the reference and we will include it in our revised paper and provide a discussion.
>
>
>
> **Concern about the regularization approximation instead of exact constraints**
>
> **Response:** We use regularization instead of constraint because its optimization is much easier and its performance is very close to that of the constrained optimization. For the constrained optimization ("strict Lovász principle"), we consider the following projection based methods:
>
> Step 1: $\hat{H}^{t} = F({A}, {X}; \theta^t)$ and ${\hat{z}}^{t} = f({A}, {X}; \phi^t)$
>
> Step 2: ${H}^{t} = \text{Proj}_U ({\hat{H}}^{t})$ and ${z}^{t} = \frac{{\hat{z}}^{t}}{\|{\hat{z}}^{t}\|}$
>
> Step 3: obtain $\theta^{t+1}, \phi^{t+1}$ by SGD updating
>
> The $\text{Proj}_U$ project ${\hat{H}}^{t}$ to the orthonormal representation space, which is similar to the Gram–Schmidt process. We define:
> $\text{proj}_w (h) := \frac{<h, w>}{<w, w>} w$.
> Let $W_k$ be the set that $W_k := \{{w}_1, {w}_2, ..., {w}_k\}.$ For each vertex $i \in V$, we denote $\Omega_i$ as the set of vector ${w}_j$s where $j$ can be each vertex not adjacent to vertex $i$. Then the ${H}^{t} = \text{Proj}_U ({\hat{H}}^{t})$ is as following:
>
> --------------------------------
>
> $ {w}_1 = \hat{h_1^t},  {e}_1 = \frac{{w}_1}{\|{w}_1\|}$
>
> ...
>
>
> $w_k = \hat{h_k^t} - \sum_{w \in W_{k-1} \cap \Omega_k} \text{proj}_w (\hat{h}_k^t),  e_k = \frac{w_k}{\|w_k\|},$
>
> ...
>
> Output ${H}^{t+1} = [{e}_1, {e}_2, ..., {e}_n]^\top$
>
> --------------------------------
>
> The comparisons between the regularized ($\mu = 1$) optimization and the constrained optimization for two methods on four datasets are as follows.
> \begin{matrix}
> \hline
> &method & MUTAG &  PROTEINS &  DD & NCI1  \\\\     \hline
>  regularized\ opt. & InfoGraph &  89.67\pm 1.54   & 75.26 \pm 1.43 &74.13\pm 1.49 &78.21\pm 1.35  \\\\
>  regularized\ opt. &GraphCL   &87.24\pm1.96    & 75.87\pm 2.17  & 79.14 \pm 1.67 &79.13\pm 1.27 \\\\
>  constrained\ opt. &InfoGraph  &86.12\pm 2.32     &75.49\pm 1.52  &76.42\pm 1.56  & 77.80\pm 1.24 \\\\
>  constrained\ opt. &GraphCL    &87.52 \pm 2.75     & 76.11\pm 1.36  & 78.54 \pm 2.21  &77.63\pm 1.58 \\\\ \hline
> \end{matrix}
> We see that the approximation has less effect on performance. We also tested the **exact penalty** method, i.e. increasing $\mu$ gradually and the performance is similar to those in the above table.
>
> **Question 1: Dealing with large-size graphs**
>
> **Response:** This is a good question. Our first formulation $\mathcal{L}_ {Lo}$ is more efficient than the Infomax principle and can handle large graphs easily. In our second formulation $\mathcal{L}_ {SLN}$, we use the truncated Lovasz kernel [8] for large-size graphs. Instead of calculating the Lovasz number for all $2^{|V|}$ vertex subsets $V$, we evaluate it on a more manageable set of subgraphs sampled from the $2^{|V|}$ possibilities. This sampling strategy significantly mitigates the computational complexity on large graphs and works very well in practice [8]. We will include the experiments on RDT-B and RDT-M5K [2] in the revised paper.
>
> [8] Johansson F et al. Global graph kernels using geometric embeddings[C] ICML 2014.
>
>
> **Question 2: Concerns about the significance of performance improvement over InfoMax principle**
>
> **Response:** Actually, the improvement especially our second method $\mathcal{L}_{ELo}$ over InfoMax principle is large in most cases. For instance, in Table 1, in terms of InfoGraph on NCI1, the performance of InfoMax principle is 76.20±1.06 while the performances of our two methods are 78.21±1.35 and 79.36±1.57 respectively. Considering that we have eight datasets, we apply the paired t-test on the mean scores over the datasets to show the significance of our methods over each of the baselines. The p-values are as follows. A p-value less than 0.05 indicates a significant difference. We see that 9 out of 10 cases are significant. This demonstrated the significance of gains given by our methods.
> \begin{matrix}
> \hline
> & InfoGraph &GraphCL &AD-GCL &JOAOv2 &AutoGCL \\\\  \hline
> InfoMax\ vs\ Lovasz& 0.00067 & 0.00286 & 0.02238 & 0.07346 & 0.00059 \\\\
> InfoMax\ vs\ Enhanced Lovasz.& 0.00005 & 0.01625 & 0.01540 & 0.01319 & 0.00035 \\\\ \hline
> \end{matrix}

---

> > ### Comment · Reviewer_JpeL · 2023-08-20
> >
> > Thank you for the clarifications. The additional experiments provided are valuable. I've reviewed the comments from the other reviewers, and it seems they didn't mention this significant related work. The distinction you made between the proposed method and existing work is also insightful. I have no further concerns at this time and will reserve any final remarks(changes) for the discussion period.

---

> > > ### Author Response · Authors · 2023-08-20
> > > **Authors' comments**
> > >
> > > Thank you for your response to our rebuttal. Please feel free to let us know if you have any questions.

---

### Author Rebuttal · Authors · 2023-08-09

* We thank the area chair and all reviewers for processing our submission. We have provided detailed responses to every reviewer. In the revision, we will fix minor issues pointed out by the reviewers and improve the presentation of this work. We will also add a comprehensive description of the data statistics, parameter settings, and starting points for each dataset in the supplementary material. We will make all of our codes publicly available.

* The attached PDF contains three figures:
   1) the intuitive example of orthonormal representation and handle vector;
   2) the flowchart (framework) of our method;
   3) the iteration performance of the loss function and its terms.

* Here we briefly summarize a few key points of our rebuttal.
   1) For Reviewer JpeL as well as Reviewer JWhm and Reviewer 2QoV, we provide some experimental of comparison between regularized optimization and constrained optimization. The results showed that the solutions are close.
   2) For Reviewer JWhm and Reviewer 2QoV, we showed the learning ability of $\mathcal{F}_W$ for solving the Lovasz problem. We defined $r=\frac{|\vartheta(G)-\vartheta(G)|}{\vartheta(G)}$ to measure the quality of $\mathcal{F}_W$. The results showed the approximation error is less than 0.1 in almost all cases. This means $\mathcal{F}_W$ can indeed learn an end-to-end solver for the Lovasz problem.
   3) For Reviewer 2QoV, we defined a metric $\ell_c(\mathcal{S}_1,\mathcal{S}_2)$ to quantify the similar or dissimilarity between graphs in the same class or in different classes. The results showed that the handle vectors are effective in providing discriminative graph-level representations.

---

### Decision · Program_Chairs · 2023-09-21

**Decision:**

Accept (poster)

**Comment:**

The reviewers mostly liked the paper from the beginning. They mention that it adds a novel perspective, is theoretically founded, and demonstrates good empirical results. The rebuttal and discussion by and large resolved the remaining issues.